# Loss of Detection of sgN Precedes Viral Abridged Replication in COVID-19-Affected Patients—A Target for SARS-CoV-2 Propagation

**DOI:** 10.3390/ijms23041941

**Published:** 2022-02-09

**Authors:** Veronica Ferrucci, Pasqualino de Antonellis, Fabrizio Quarantelli, Fatemeh Asadzadeh, Francesca Bibbò, Roberto Siciliano, Carmen Sorice, Ida Pisano, Barbara Izzo, Carmela Di Domenico, Angelo Boccia, Maria Vargas, Biancamaria Pierri, Maurizio Viscardi, Sergio Brandi, Giovanna Fusco, Pellegrino Cerino, Livia De Pietro, Ciro Furfaro, Leonardo Antonio Napolitano, Giovanni Paolella, Lidia Festa, Stefania Marzinotto, Maria Concetta Conte, Ivan Gentile, Giuseppe Servillo, Francesco Curcio, Tiziana de Cristofaro, Francesco Broccolo, Ettore Capoluongo, Massimo Zollo

**Affiliations:** 1CEINGE Biotecnologie Avanzate, 80145 Naples, Italy; veronica.ferrucci@unina.it (V.F.); deantonellis@ceinge.unina.it (P.d.A.); asadzadeh@ceinge.unina.it (F.A.); bibbo@ceinge.unina.it (F.B.); sicilianor@ceinge.unina.it (R.S.); sorice@ceinge.unina.it (C.S.); pisano@ceinge.unina.it (I.P.); barbara.izzo@unina.it (B.I.); didomenico@ceinge.unina.it (C.D.D.); boccia@ceinge.unina.it (A.B.); giovanni.paolella@unina.it (G.P.); ettoredomenico.capoluongo@unina.it (E.C.); 2Dipartimento di Medicina Molecolare e Biotecnologie Mediche (DMMBM), University of Naples Federico II, 80131 Naples, Italy; quarantelli@ceinge.unina.it; 3Department of Neurosciences, Reproductive Sciences and Odontostomatology, University of Naples Federico II, 80131 Naples, Italy; maria.vargas@unina.it (M.V.); giuseppe.servillo@unina.it (G.S.); 4Istituto Zooprofilattico Sperimentale del Mezzogiorno, 80055 Naples, Italy; biancamaria.pierri@izsmportici.it (B.P.); maurizio.viscardi@cert.izsmportici.it (M.V.); sergio.brandi@cert.izsmportici.it (S.B.); giovanna.fusco@izsmportici.it (G.F.); strategia@izsmportici.it (P.C.); 5U.O.C. Pneumologia, Ospedale Boscotrecase—Asl Napoli3 Sud, 80042 Naples, Italy; depietro.livia@libero.it; 6Dipartimento di Medicina di Laboratorio e Anatomia Patologica, Nola Hospital, Asl Napoli 3 Sud, 80035 Naples, Italy; cifurfar@tin.it (C.F.); la.napolitano@aslnapoli3sud.it (L.A.N.); 7Unità Speciale di Continuità Assistenziale (USCA), Asl Benevento, 82100 Benevento, Italy; festalidia61@gmail.com (L.F.); mc.conte1@virgilio.it (M.C.C.); 8Dipartimento di Medicina Clinica e Chirurgia, University of Naples Federico II, 80131 Naples, Italy; ivan.gentile@unina.it; 9Dipartimento di Medicina di Laboratorio, Università degli Studi di Udine, 33100 Udine, Italy; stefania.marzinotto@asufc.sanita.fvg.it (S.M.); francesco.curcio@uniud.it (F.C.); 10IEOS-Institute of Experimental Endocrinology and Oncology G. Salvatore, National Research Council, 80131 Naples, Italy; t.decristofaro@ieos.cnr.it; 11BioMol Laboratories srl, 80146 Naples, Italy; 12Laboratory of Molecular Microbiology and Virology, School of Medicine, University of Milano-Bicocca, 20900 Monza, Italy; broccolof@gmail.com; 13DAI Medicina di Laboratorio e Trasfusionale, University of Naples Federico II, 80131 Naples, Italy

**Keywords:** SARS-CoV-2 virus particles, qPCR methods, 2′-*O*-methyl antisense RNA, sgN, sgE

## Abstract

The development of prophylactic agents against the SARS-CoV-2 virus is a public health priority in the search for new surrogate markers of active virus replication. Early detection markers are needed to follow disease progression and foresee patient negativization. Subgenomic RNA transcripts (with a focus on sgN) were evaluated in oro/nasopharyngeal swabs from COVID-19-affected patients with an analysis of 315 positive samples using qPCR technology. Cut-off Cq values for sgN (Cq < 33.15) and sgE (Cq < 34.06) showed correlations to high viral loads. The specific loss of sgN in home-isolated and hospitalized COVID-19-positive patients indicated negativization of patient condition, 3–7 days from the first swab, respectively. A new detection kit for sgN, gene E, gene ORF1ab, and gene RNAse P was developed recently. In addition, in vitro studies have shown that 2’-*O*-methyl antisense RNA (related to the sgN sequence) can impair SARS-CoV-2 N protein synthesis, viral replication, and syncytia formation in human cells (i.e., HEK-293T cells overexpressing ACE2) upon infection with VOC Alpha (B.1.1.7)-SARS-CoV-2 variant, defining the use that this procedure might have for future therapeutic actions against SARS-CoV-2.

## 1. Introduction

SARS-CoV-2 is the human coronavirus (CoV) responsible for the CoV disease 19 (COVID-19) pandemic. Human CoVs are members of the Nidovirales order and belong to the Coronaviridae family. To date, seven species of human CoVs have been described: HCoV-NL63, HCoV-229E, HCoV-OC43, HCoVHKU1, SARS-CoV, MERS-CoV, and SARS-CoV-2. Like other CoVs, SARS-CoV-2 is an enveloped virus with a positive-sense, single-stranded RNA genome. SARS-CoV-2 belongs to the genus betacoronavirus, together with SARS-CoV and MERS-CoV (with 80% and 50% identity, respectively) [1].

Coronaviruses, including SARS-CoV-2, have the largest genomes (26–32 kb) among all of the RNA virus families, which are flanked by 5′ and 3′ untranslated regions. SARS-CoV-2 RNA contains a common ‘leader’ sequence (of 70 nt) [2]. Upon cell entry, the viral genomic RNA (gRNA) is translated to produce nonstructural proteins from two large open reading frames (ORFs), ORF1a, and ORF1b, via proteolytic cleavage. Among these, 15 nonstructural proteins make up the viral replication and transcription complex [3]. Of importance, Nsp12 (which harbors RNA-dependent RNA polymerase; RdRp) leads the viral replication and transcription mechanisms by using viral RNA as the template.

A hallmark of CoVs is a “discontinuous transcription” mechanism that produces a set of subgenomic RNAs (sgRNAs). Indeed, during their viral cycle, CoVs replicate their genomic RNA to produce full-length negative-sense RNA molecules that act as the templates for the synthesis of positive-sense gRNAs that are then packaged by the structural proteins into newly assembled virions. However, the ORFs are transcribed from the 3′ one-third of the genome to form sgRNAs that encode the SARS-CoV-2 structural proteins (i.e., spike [S], envelope [E], membrane [M], nucleocapsid [N]), and several accessory proteins (e.g., 3a, 6, 7a, 7b, 8, 10), according to the “leader-to-body fusion” model [1]. Briefly, during the negative-strand synthesis, the viral replication and transcription complex interrupts transcription when it crosses a transcription regulatory sequence (TRS) upstream of most ORFs in the 3′ one-third of the viral genome (i.e., a TRS ‘body’ or TRS-B). The synthesis of the negative-strand RNA is then re-initiated at the TRS in the leader sequence (TRS-L) at 70 nucleotides from the 5′ end of the genome, due to the interaction between TRS-B of the negative-sense nascent RNA and TRS-L of the positive-sense gRNA. Upon re-initiation of RNA synthesis at the TRS-L region, a negative-strand copy of the leader sequence is added to the nascent RNA to complete the synthesis of negative-strand sgRNAs. These fused negative-strand intermediates are used as templates to synthesize positive-sense sgRNAs that are translated into both structural and accessory proteins [3]. In addition, in a global landscape analysis of SARS-CoV-2 subgenome RNA expression, Wang et al., 2021 [4] used computational analysis to identify novel modes of viral sgRNA biogenesis via a ‘TRS-independent’ mechanism.

To date, eight main sgRNAs have been reported to be produced in SARS-CoV-2-infected cells. In addition to these canonical sgRNAs, noncanonical RNA products of discontinuous transcription have also been reported for SARS-CoV-2, including fusions of the 5′ leader sequence to unexpected 3′ sites, TRS-L-independent long-distance fusions, and local fusions that result in small deletions mainly in structural and accessory genes [1].

Of interest, the N protein sgRNA (sgN; which codes for the N protein) is the most abundant sgRNA during viral infection, mostly due to the low DG value in the duplex formation between TRS-L and TRS-B [2]. On this basis, of all the sgRNAs, sgN has been shown to be the most abundant in SARS-CoV-2-infected cells [5]. Furthermore, together with sgRNA for Orf7a, the same sgN also shows highest abundance in swab samples from COVID-19-affected patients [6]. Another important finding related to the SARS-CoV-2 N protein encoded by the sgN transcript is its role in the regulation of the discontinuous transcription process through its C-terminal domain, which retains its interaction with TRS sequences, and the consequential regulation of transcription [7]. To this end, early and continuous expression of the sgN transcript ensures the generation of the N protein in multiple copies in nascent viral genome particles, which is of great importance for SARS-CoV-2 replication and propagation.

As viral sgRNAs are transcribed in infected cells but are not packaged together with gRNAs into nascent virions, they might be useful indicators of the presence of active infection [8]. However, to date, the literature data are contradictory on which sgRNAs might represent indicators of viral status. The detection of sgRNA for the E gene (i.e., sgE) has been shown in oro-pharyngeal throat swabs samples collected from days 4 to 9 after COVID-19 symptom onset. The authors would thus suggest that sgE can be used as an indicator of active SARS-CoV-2 infection [9]. An additional study reported that the detection of sgRNAs was possible up to 11 and 17 days after first detection of SARS-CoV-2 infection through PCR and next-generation sequencing, respectively [6]. Therefore, this study concluded that sgRNAs cannot be used as indicators of active CoV replication/infection, arguing that this might reflect the methodology used. Indeed, to date, the RNAs evaluated in diagnostic swab samples are likely to be a mixture of both gRNAs and sgRNAs. Alexandersen et al. [6] concluded that these sgRNAs are protected from nuclease actions and hydrolysis by “double-membrane vesicles”, and they concluded that this is the main reason why sgRNAs survive longer in cells. In another study performed on in vivo infection of rhesus macaques, Dagotto et al. [10] investigated whether an sgRNA assay can distinguish an input challenge virus from an actively replicating virus in vivo, through a comparison of the expression of both sgE and the standard N and E proteins (both at the level of gRNA and sgRNA detection). In summary, they suggested the use of sgRNAs to monitor the actively replicating virus in prophylactic and therapeutic SARS-CoV-2 studies during rhesus macaques’ infection. In our previous study, we showed that sgN detection might provide a better candidate biomarker for active and higher viral loads in SARS-CoV-2-infected patients than sgE. Of interest, sgN expression was not influenced by the expression of genomic transcription of the N gene [11]. In a recent study, Oranger et al., 2021 [12] used fine-tuned droplet digital PCR to show that both sgN and sgS directly correlate to gRNA copies, and then that the sgN and sgS expression levels are reduced in RNA samples with low viral RNA content. This thus indicated that the samples analyzed were mainly characterized by residual genomic SARS-CoV-2 material with little or no active viral transcription.

On the basis of our previous results, to determine whether sgN detection can be used as a marker of active viral replication, we undertook a multicenter study here across five Diagnostic Coronet centers in Italy, which included 315 oro/nasopharyngeal swabs of COVID-19-positive patients. In this report, we define the limit of detection of sgN, sgE, and Orf1b, and we underline that this thus precedes patient negativization in home-isolated and hospitalized COVID-19-positive patients (by ~3/7 days from the first swab, respectively). We have also designed a new kit for detection of sgN, gene E, gene ORF1ab, and RNAse P gene that can detect their levels in oro/nasopharyngeal swabs and bronchial aspirates samples. SARS-CoV-2 life-cycle has been previously reported to trigger cell–cell fusion mechanisms, thus orchestrating the formation of multinucleated cells [13,14]. We then show that targeting the sgN sequence 2’-*O*-methyl antisense RNA can impair viral replication and syncytia formation in human cells (i.e., HEK-293T cells overexpressing ACE2) upon infection with the VOC Alpha B.1.1.7-SARS-CoV-2 variant. Sequence analysis of the VOC variants shows that the sequences match to 100% identity to Beta and Gamma and 97.3% to Omicron, thus enhancing our hopes and findings. Altogether, these analyses indicate future therapeutic implications to target N protein synthesis to inhibit SARS-CoV-2 viral replication.

## 2. Results

### 2.1. sgN and sgE Are Detected in Samples from COVID-19-Affected Patients with High SARS-CoV-2 Viral Load

Recently, we reported that in a small cohort of COVID-19-affected patients, sgN expression could only be detected in samples with high viral load [11]. Here, we have developed a diagnostic kit, named “SARS-CoV-2 Viral3”, based on a Taqman approach (BioMol laboratories; https://www.biomollaboratories.it/, accessed on 30 December 2021) that can detect expression levels of viral sgN, gene E, gene ORF1ab, and the human RNAse P gene. We compared the results obtained from 50 oro/nasopharyngeal swabs to those obtained using the “in vitro diagnostic” (IVD) approved Allplex 2019-nCoV assay (Seegene; https://www.seegene.com/, accessed on 30 December 2021). These data show that these kits can identify with certainty the SARS-CoV-2-positive patients (Appendix A). Furthermore, we demonstrate that the new SARS-CoV-2 Viral3 kit can identify ‘true negative’ COVID-19-free people through analysis of an independent cohort of 12 samples (Appendix A).

The SARS-CoV-2 Viral3 kit also identified viral sgN, gene E, and gene ORF1ab in SARS-CoV-2-positive bronchial aspirate specimens collected from hospitalized patients, with a comparison of oro/nasopharyngeal swabs and bronchial aspirate specimens presented in Appendix A.

The SARS-CoV-2 Viral3 kit was also evaluated for sensitivity (sgN, gene E, gene ORF1ab; 300,000 to 30 viral copies) and for sgN specificity (≥99.9%), with a hit rate of 95.0% (see Appendix A). We tested the detection of SARS-CoV-2 sgN transcripts using the SARS-CoV-2 Viral3 kit through the analysis of oro/nasopharyngeal swab samples from a cohort of 315 COVID-19-positive Italian patients (in Coronet Laboratories based in Milan, Udine, Naples, Italy). The positivity of these patients to SARS-CoV-2 infection was confirmed through detection of viral gene E and gene ORF1ab in all the samples. In contrast, the levels of sgN were not detectable (i.e., Cq > 40) in 120 of these samples (38.1%) (Appendix A). One-way analysis of variance (ANOVA) was used to determine that sgN expression was detected using the SARS-CoV-2 Viral3 kit only in the samples that were characterized by Cq values < 33.163 for viral gene E (*p* < 0.0001; Figure 1A) and <33.155 for gene ORF1ab (*p* < 0.0001; Figure 1B). Furthermore, as expected, the expression levels of the human RNAse *p* gene did not influence the detection of viral sgN (*p* < 0.8; Appendix A). Altogether, these data confirm that expression of viral sgN was detected using the SARS-CoV-2 Viral3 kit only in those samples with higher viral loads, thus confirming our previous findings [11] in this additional large cohort analysis.

As the detection of sgE has also been suggested as an indicator of active SARS-CoV-2 infection [9], we also compared sgE expression levels (using Taqman methodology) to sgN expression levels in 122 patients from one of the single Coronet centers, as part of the full cohort (ASL Napoli3-sud; Appendix A). These data showed that sgN and sgE were not detectable using the SARS-CoV-2 Viral3 kit in terms of their levels of expression in 82.8% and 64.8% of this single-center cohort, respectively (Figure 1C,D). An ANOVA was again used to determine that sgN expression was detected using the SARS-CoV-2 Viral3 kit only in those patients with Cq values <33.4 for viral gene E (*p* < 0.0001; Figure 1E) (gene E cut-off as the limit of detection) and <33.54 for gene ORF1ab (*p* < 0.0001; Figure 1G) (gene ORF1ab cut-off for the limit of detection). These analyses also showed similar results for the entire cohort (i.e., 315 samples; Figure 1A,B). For the expression of sgE levels, detection was seen using the SARS-CoV-2 Viral3 kit when the Cq values for viral gene E and gene ORF1ab were <34.06 and <34.20, respectively (*p* < 0.001, for both; Figure 1F,H).

Taken together, these data indicated that both of the sgRNA transcripts (i.e., sgN, sgE) are independently detected only in those patients with higher viral loads, when the infection is expanding and rapidly progressing. Vice-versa, at lower viral loads, sgN was generally not detected (gene E Cq >33.16; gene Orf1b Cq >33.15; see Figure 1A,B). Similarly, considering the lower viral loads, the expression of sgE was not detectable (gene E Cq > 34.06; gene ORF1ab Cq > 34.2; see Figure 1F,H). These new sets of data demonstrated differences from our previous report [11], which we believe is because of the improved sensitivity of the SARS-CoV-2 Viral3 kit and the increased number of samples (48 versus 315 samples) in the analysis here.

### 2.2. Loss of Detection of sgN Precedes SARS-CoV-2 Replication Failure in Home-Isolated and Hospitalized COVID-19-Affected Patients

With the aim of monitoring viral replication and its potential failure, we undertook further analyses to answer the question of how longitudinal expression occurs for the sgRNAs (i.e., sgN, sgE) and for the genes N, E, ORF1ab, and RpRd that are expressed during SARS-CoV-2 infection. Here, we analyzed a cohort of oro/nasopharyngeal swabs collected from 16 COVID-19-positive home-isolated patients at specific times (i.e., 3-day intervals from the first swab) until they reached a negative status for the SARS-CoV-2 genes, when possible (Figure 2A, Appendix A). Among these 16 patients, 10 were followed up to 7 days from the first swab analysis, and the remaining 6 patients to 6 days (Figure 2A, Appendix A). We used both kits (i.e., SARS-CoV-2 Viral3 assay, Allplex 2019-nCoV assay) with these oro/nasopharyngeal swabs to determine the levels of viral sgN, gene E, gene ORF1ab, gene N, and human RNase P gene. Three days and 7 days from the first swab, sgN was detected in 44% and 10% of the patients, respectively (Figure 2B and Appendix A). In contrast, loss of detection of the other viral genes was seen for patients 7 days from the first swab test (detected in: gene E, 20%; gene ORF1ab, 20%; gene N, 30%; RdRp 10%) (Figure 2B and Appendix A). In the same cohort analysis, sgE was detected in 100% of the patients after 3 days, and in only 13% after 7 days (Figure 2B and Appendix A) while the other genes analyzed here were detected at the same levels as discussed above.

In more detail, gene ORF1ab and RdRp were detected only in 10% of the patients 7 days from the first swab (Figure 2B and Appendix A). Similarly, genes E and N were detected in 20% and 30%, respectively, of the patients 7 days from the first swab (Figure 2B and Appendix A). Several reasons can be ascribed to the loss of sgN detection, and the most appropriate could be due to its rapid degradation compared to other viral markers. However, literature data show that sgRNAs are protected through “double-membrane vesicles” from the hydrolytic actions of intracellular nucleases (see Alexandersen at al. [6]). Moreover, due to the specificity (≥99.9%) and sensitivity (1.5 copies/mL for Allplex 2019-nCoV and 30 viral copies/mL for SARS-CoV-2 Viral3, see Appendix A) of the commercial kit used here, it is unlikely that only sgN has a higher rate of degradation compared to the other markers (Allplex 2019-nCoV: gene N, gene E, gene RpRd; SARS-CoV-2 Viral3: sgN, gene E, gene ORF1ab). Furthermore, we have previously reported a direct correlation between sgN expression and the viral load (MOI) in SARS-CoV-2-infected Vero E6 cells at different viral particle numbers [11]. However, at this time, we cannot exclude an additional hypothesis of a potential higher rate of instability and less efficient RT-PCR detection for sgN RNA in comparison to the other viral genomic and subgenomic targets.

We then analyzed an independent cohort of six COVID-19-affected patients hospitalized in an Intensive Care Unit. Here, the analysis monitored sgN, sgE, gene E, gene ORF1ab, gene N, and RdRp using longitudinal detection at 7-day intervals (0, 7, 14 days; Appendix A). SgN was detected on the first swab tests, and again in 67% of the patients after 7 days, and in 33% after 14 days (Figure 2C,D and Appendix A). Then there was loss of detection of the other viral genes in these patients after 7 days and 14 days (detected in, respectively: sgE 80%, 33%; gene E, 83%, 50%; gene ORF1ab, 83%, 50%; gene N, 67%, 50%; RdRp, 67%, 50%).

Taken together, these data obtained through the analysis of two independent datasets of oro-pharyngeal swab tests from COVID-19-affected patients (home-isolated, hospitalized) identified sgN as the first viral transcript to show decreased expression levels (to the ‘undetectable’ level) during their recovery period of SARS-CoV-2 infection. Altogether these data also show that the detection of sgN in these two independent COVID-19 patient cohorts match with the viral E and Orf1ab cut-off for the limit of detection (showed in Figure 1A,B) in 81.8% of the cases analyzed (Appendix A, and Appendix A). Overall, sgN detection preceded the benign SARS-CoV-2 negativization by 3 to 7 days from the first swab in home-isolated and hospitalized COVID-19-positive patients, respectively.

### 2.3. Reduction of Viral Load by Targeting sgN in HEK293T-ACE2 SARS-CoV-2–Infected Cells

SARS-CoV-2 has now been reported as inducing systemic perturbations by impacting multiple organs. In this regard, the lung is the primary target for SARS-CoV-2 that causes an early respiratory infection. Then, mostly due to the wide expression of ACE2 in a heterogeneous population of systemic cells, SARS-CoV-2 can damage several systemic tissues and result in multi-organ dysfunction, including kidney due to high ACE2 expression levels [15]. HEK-293T cells have been widely used as a cellular model to identify seral mechanisms of action. Here, in order to allow and normalize SARS-CoV-2 infection, we have generated HEK-293T stable cell clones overexpressing human ACE2 cDNA under the control of CMV promoter (Appendix A). This cellular model overcomes those alterations in the viral infection efficiency due to the different multiplicity of infection (MOI). Furthermore, at this time, kidney cells (including HEK-293 expressing ACE2) are used for in vitro model platforms for SARS-CoV-2 infection [16,17].

In this regard, we first generated the HEK293T-ACE2 cellular model by creating stable clones using a PCDNA3.1 +C-DYK (Addgene vector name: pCEP4-myc-ACE2: containing ACE2 cDNA human wild-type). After antibiotic selection for a hygromycin resistance gene, several clones were selected. One clone used here is shown in Appendix A, which contains a high copy number endogenous plasmid that expressed the ACE2 protein (2000-fold compared to HEK293T wild-type cells), with expression of a significant amount of the protein in the membranes of these cells (Appendix A).

As sgN is the most abundant sgRNA during SARS-CoV-2 infection, and due to its pivotal role in viral genome packaging, we set-out here to transiently transfer into the host cells (i.e., HEK293T-ACE2) an oligonucleotide as antisense against the TRS sequence linked to the first ATG corresponding to the most 5′ end of the of N gene (following the leader sequence). We then infected these cells with VOC Alpha (B1.1.7) SARS-CoV-2 virus, to thus measure sgN, E, N, and sgE with the intent of quantifying their virus load. For this purpose, we used sense and antisense 2’-*O*-methyl RNA oligonucleotides that targeted part of the leader sequence of SARS-CoV-2 followed by TRS and the 5′ end of gene N (Figure 3A, yellow arrow with red dashed lines). This sequence was confirmed by Sanger sequence analysis from a swab sample from a COVID-19-affected patient (Appendix A). We designed the sequence after aligning it with all the SARS-CoV-2 variants identified to date. Figure 3B shows the sequence alignment with the nucleotides that comprise the designed 2’-*O*-methyl RNAs aligned with each independent VOC variant showing nucleotide variations. The sequence identities seen were 81.1% for Alpha, 100% for Beta, 100% for Gamma, 89.2% for Delta, and 97.3% for Omicron (see Figure 3B and Appendix A). As additional help to draw the oligonucleotide sequence, we obtained sequencing data from an RNA Nanopore sequencing approach using Vero E6 monkey kidney cells previously infected with the 20A clade SARS-CoV-2 virus [5]. These analyses allowed the determination of the best match sequence junction between the leader and TRS-B sequences of sgN (see Figure 3A,B). Of note, the sequence was also confirmed with 100% identity in the sgN sequence with a frequency of 82.2% on a total 5781 sgN sequences obtained by the Nanopore data (VeroE6 cells infected with SARS-CoV-2- 20A at 0.1 MOI for 60 h of infection). These sequences have been deposited at Sequence Read Archive SRA–BioProject PRJNA688696, see additional data [5].

The synthesis and purification of the 2′-O-methyl sense or antisense RNA nucleotide was carried out following the procedures described in Material and Methods. We then transfected 2′-*O*-methyl sense or antisense RNA (1 mg) into HEK-293T cell clones stably overexpressing human ACE2 (i.e., HEK293T-ACE2 cells; Figure 3C). Twelve hours from the start of transfection, the cells were infected with B.1.1.7 VOC Alpha SARS-CoV-2 viral particles (0.03 MOI) for a further 36 h (Figure 3C). Of importance, these virus particles contained the virus RNA sequence (sgN) that shows 81.1% identity to the sequence of the 2′-*O*-methyl antisense RNA. Immunoblotting analysis showed a statistically significant decrease in viral N protein levels in these SARS-CoV-2-infected cells previously transfected with 2′-*O*-methyl antisense RNA (Figure 3D). To ascribe the reduced N protein levels to the impairment of sgN transcripts, we also performed qPCR analysis using the SYBR green and Taqman methodologies [5] (see Material and Methods) to detect the viral gene transcripts. The data presented in Figure 3E show specific reduction of sgN levels by the 2′-*O*-methyl antisense RNA. This thus confirmed that although there was 81.1% identity between the 2’-*O*-methyl antisense RNA and the sgN antisense technology, this percentage identity was sufficient to impair sgN translation (Figure 3E). As a consequence, the levels of other SARS-CoV-2 genes (i.e., gene s E, N1-3) were also decreased (Figure 3E) only in the infected cells that expressed 2’-*O*-methyl antisense RNA. At this time, this phenomenon can be ascribed to: (i) a reduction in the N protein, which has a role in the discontinuous transcription mechanism; and (ii) a substantial reduction in virus replication after 36 h of infection.

### 2.4. Reduction of Syncytia Formation by Targeting sgN in HEK293T-ACE2 SARS-CoV-2–Infected Cells

Syncytia phenomena are related to the persistence of the viral RNA infection and replication in SARS-CoV-2-infected patients [13,18,19]. Syncytia can also be induced by certain types of infections by viruses, such as human immunodeficiency virus, respiratory syncytial virus, and herpes simplex virus [20]. SARS-CoV-2 virus-induced cell fusion facilitates the transfer of viral genomes to neighboring cells. In lymphocytes in the human lung, widespread syncytia formation has been seen between SARS-CoV-2-infected pneumocytes and healthy pneumocytes in patients infected with SARS-CoV-2 [21]. A model through ACE2 and S protein interaction was described showing how syncytia can be used to facilitate virus transmission. However, the viral and cellular mechanisms that regulate the formation of syncytia during SARS-CoV-2 infection remains largely elusive. We asked here whether syncytia formation was inhibited by transfection with 2′-O-methyl antisense RNA, as the cellular model of infection (HEK293T-ACE2 cells), mainly because the S protein is very low in cells infected by SARS-CoV-2, and this would affect syncytia formation by impairing the ACE2-S protein interactions that prime syncytia formation. Indeed, upon SARS-CoV-2 infection, the cells that received 2′-*O*-methyl antisense RNA showed less S protein expression, and, as a consequence, less syncytia formation, as revealed by high-resolution confocal microscopy (Figure 4A). The additional measure of proportion of syncytia further demonstrated this impaired syncytia formation (*p* < 0.001 versus the control; *p* < 0.01 versus the sense 2′-*O*-methyl RNA treated cells; see Figure 4B). Altogether, these data show that the biogenesis of sgN goes through a “leader to TRS body fusion model” that produces sufficient copies of new sgN mRNAs to translate them into N proteins that are used by the gRNA to be packaged into nascent virions. We show here that in vitro it is possible to impair this process using 2′-*O*-methyl antisense RNA oligos, with therapeutic benefits seen through reduction of viral load in SARS-CoV-2-infected cells, thus further inhibiting syncytia formation and subsequent virus propagation (Figure 5A). Then the model describes that during SARS-CoV-2 infections, the acute phase of virus replication is enhanced by expression of viral genome and subgenomic transcripts (Figure 5B); while during the negativization process, the loss of sgN detection is seen (i.e., undetectable levels: Cq > 40; viral ORF1ab and E Cq values > 33.15 and 33.16, respectively).

## 3. Discussion

SgRNAs are only produced during active infection to generate the N protein units to cover the nascent SARS-CoV-2 RNA genome that will be encapsulated in new virus particles, and that thus presents an accurate measure of replicating virus. The N protein, which is the only protein present in the coronavirus nucleocapsid, plays a critical role in ensuring coronavirus replication and successful intracellular lifecycle, thus it is considered a suitable target when designing new vaccines together with S-RBD protein [22,23]. At this time, a question can be raised on why do we observe a loss of sgN before the other markers during the time of clinical virus negativization? We thus think that this is due to the importance of sgN on supplying N protein for generating intact new genome RNA copies of the viral particles once assembled, hence underlying its functional importance on sustaining the viral stability and potency. In this case, missing N protein, as observed on measuring sgN RNA copies loss, supports the hypothesis that virus replication and infection capability is diminishing as a sign of negativization. At this time, we cannot exclude that sgN RNA, although present in a larger copy number during viral RNA genome replication (see data presented by Nanopore sequencing, BioProject PRJNA688696, [5]) would be more sensible to RNA instability in comparison with other target viral mRNA genes. Future laboratory settings will address these hypotheses. The routine diagnostic tools based on RT-PCR assays typically target total or genomic SARS-CoV-2 RNA, and are thus not an optimal measure of newly replicating active virus. Here, we have developed a new Taqman-based diagnostic assay that can detect the expression of the SARS-CoV-2 sgN transcript together with viral gene E, gene ORF1ab, and human RNAse P gene. These data demonstrate the potential of measuring sgN transcripts rather than gRNA as a more specific measure of the replicating virus in samples with higher viral load. In this regard, sgN was not detectable in oro/nasopharyngeal swabs from COVID-19-affected patients showing Cq values >33.163 for viral gene E (*p* < 0.0001; Figure 1A), and >33.155 for gene ORF1ab (*p* < 0.0001; Figure 1B) genes. Of importance, this method has also been validated here in bronchial aspirate specimens (Appendix A).

We further shown the ‘prognostic’ value of sgN detection in hospitalized (Intensive Care Unit) and home-isolated COVID-19-positive patients. Indeed, the SARS-CoV-2 Viral3 diagnostic kit was compared with the commercially available Allplex 2019-nCoV assay to monitor the time to reach negative results for the SARS-CoV-2 biomarkers, as a sign of benign disease and recovery from disease (follow-up: 3–7 days home isolated, 1–2 weeks hospitalized patients) through detection of viral sgN, gene E, gene ORF1ab, gene N, and human RNase P gene. The data presented here show that sgN is the first transcript that becomes undetectable during the recovery in both hospitalized and isolated COVID-19-affected patients (Figure 2). These results suggest that sgN loss can be considered as a ‘predictive marker’ for lower SARS-CoV-2 replication activity, thus being of importance for both monitoring the therapeutic response and alerting clinicians that the SARS-CoV-2 negativization processes is underway.

Developing prophylactic actions for the SARS-CoV-2 virus is a public health priority. One of the most important actions in the diagnostic SARS-CoV-2 setting is to define when the diminishing viral load can be considered a benign phase (i.e., viral negativization). Our method of detection of sgN has a specificity of ≥99.9% and sensitivity for sgN, gene E, and gene ORF1ab of 300,000 to 30 viral copies, with hit rate of 95%. This positions the SARS-CoV-2 Viral3 kit as a valuable tool for early detection and negativization, and consequently to estimate the infectiousness of SARS-CoV-2 patients world-wide.

Viral structural proteins are essential for virus survival and propagation. These SARS-CoV-2 structural proteins (i.e., S, E, M, N proteins) are encoded by the SARS-CoV-2 sgRNAs that are also responsible for the synthesis of several other accessory proteins (i.e., 3a, 6, 7a, 7b, 8, 10). N protein encapsulates viral genomic RNAs during the viral life cycle to protect the genome and co-enter the host cell with the viral genomic RNAs, which indicates that N is essential for viral RNA replication, particularly at the initiation stage. SgRNAs are expressed in abundance [4,6], and among them, sgN has been shown to be the most abundant in SARS-CoV-2-infected cells [1,5]. To date, a siRNA-based approach that targets the leader sequence of SARS-CoV [24] and a chemical inhibitor that targets the RNA-binding affinity of N protein [25] have been tested in vitro only against SARS-CoV infection. The effectiveness of these therapeutic approaches against SARS-CoV-2 have not been tested to date. Further, there are no prophylactic treatments available. Here, we used sense and antisense 2′-O-methyl RNA oligonucleotides that specifically target the TRS sequence (following the leader sequence) at the 5′ end of the of N gene in B.1.1.7 SARS-CoV-2-infected HEK-293T cells overexpressing ACE2. Through qPCR assays, immunoblotting, and high-resolution immunofluorescence, we have shown that sense 2’-*O*-methyl RNA oligonucleotide reduces sgN transcripts, and, as a consequence, N protein levels, and therefore inhibits SARS-CoV-2 infection and replication. Furthermore, N protein has been shown to have a critical role during the discontinuous transcription process as a positive regulator that influences the expression of the other sgRNAs (Yang et al. 2021). The results presented here indeed show that lowering the expression of sgN results in inhibition of the gRNA E and sgRNA E transcripts (Figure 3E), which probably occurs via competition for the viral replication machinery.

In the future, it would appear reasonable to explore other sgRNAs in similar in vitro therapeutic assays. Of importance, we envision the use of technologies based on nanoparticles and stable nucleic acid lipid particles for delivery of 2′-*O*-methyl antisense RNA in human cells infected with SARS-CoV-2 [26], as proof-of-principle of the use of therapeutic nanoparticles. Together with an aerosol application formula, this would be of great value for the treatment of patients with respiratory failure [5], and especially for individuals who cannot mount an immune response to the vaccines (e.g., immunosuppressed, under transplants) and thus requiring a prophylactic to prevent infection after being exposed to SARS-CoV-2. Additionally, by analyzing the 2′-*O*-methyl antisense RNA sequence, we found 100% identity to VOC Beta and Gamma, 91.8% to VOC Delta, 97.3% to VOC Omicron, and 81.0% to VOC Alpha (Figure 3B). This thus suggests further that 2′-*O*-methyl antisense RNA can also act against the other VOC variants with a greater percentage identity as the VOC Alpha used here (Figure 3C–E and Figure 4A,B). Further studies should address its therapeutic usefulness for the other VOC variants.

Nevertheless, we should consider at this time that the canonical “leader to body fusion model” is not unique, and that others have been proposed (e.g., TRS-independent mechanism, multi-switch sgRNA synthesis), and thus other 2′-*O*-methyl antisense RNAs with different switch junctions can be designed, as identified by [4,27].

One of the most common phenomena in SARS-CoV-2 infection is related to syncytia formation [17,28]. Syncytia can also be induced by certain types of infections by viruses, such as human immunodeficiency virus, respiratory syncytial virus, and herpes simplex virus [20]. One of the most accredited models of syncytia formation is that this virus-induced cell fusion serves to facilitate the transfer of viral genomes to the neighboring cells, thus enhancing viral propagation [21]. Here, we showed significant inhibition of syncytia formation (*p* < 0.01) in those cells overexpressing the 2′-*O*-methyl antisense RNA sgN sequence using SARS-CoV-2 VOC Alpha infected human cells (Figure 4A). This is a further evidence that inhibition of N protein translation, and consequentially the transcription of the other proteins (e.g., S protein) [7], negatively affects viral replication via the syncytia model of action.

## 4. Materials and Methods

### 4.1. Cell Culture

HEK-293T cells (CRL-3216, ATCC, Manassas, Virginia, USA), HEK-293T stable clones overexpressing human ACE2 (HEK293T-ACE2) and Vero E6 cells (C1008, ATCC) were grown in a humidified 37 °C incubator with 5% CO_2_. The cells were cultured in feeder-free conditions using Dulbecco’s modified Eagle’s medium (41966-029; Gibco, Thermo Fisher Scientific, Waltham, MA, USA) with 10% fetal bovine serum (10270-106; Gibco), 2 mM L-glutamine (25030-024; Gibco), and 1% penicillin/streptomycin (P0781; Sigma-Aldrich), with medium changed daily. Cells were dissociated with Trypsin-EDTA solution (T4049, Sigma-Aldrich, St. Louis, MO, USA) when the culture reached ~80% confluency.

### 4.2. Generation of HEK293T-ACE2 Stable Clones

HEK-293T cells were plated in 6-well plates in 2 mL of Dulbecco’s modified Eagle’s medium (41966-029; Gibco) with 10% fetal bovine serum (10270-106; Gibco). When the culture reached ~70% confluency, they were transfected with pCEP4-myc-ACE2 plasmid (#141185, Addgene) with X-tremeGENE 9 DNA Transfection Reagent (06365779001; Sigma-Aldrich). Briefly, X-tremeGENE 9 DNA Transfection Reagent was equilibrated at room temperature (+15 to +25 °C) and diluted with serum-free Dulbecco’s modified Eagle’s medium (41966-029; Gibco) to a concentration of 3 μL reagent/100 μL medium. Then, 1 μg of DNA plasmid was added to 100 μL of diluted X-tremeGENE 9 DNA Transfection Reagent (3:1 ratio [μL]). The transfection reagent:DNA complex was then incubated for 15 min at room temperature. Finally, the transfection complex was added to the cells in a dropwise manner. Following forty-eight hours from transfection, the cell culture medium was changed, and the cells’ clones were selected using 800 μg/mL hygromycin.

### 4.3. Transient Transfection with 2′-O-Methyl RNA Oligos Targeting sgN

#### 4.3.1. 2′-O-Methyl RNA Oligos Design

The following 2’-*O*-methyl RNA oligonucleotides were designed against the junction site between the leader sequence, the transcription-regulating sequences (TRSs) and the 5′ end of gene N (leader-“junction”-TRS-“junction”-N) of SARS-CoV-2:

Sense: TCTCTAAACGAACAAACTAAAATGTCTGATAATGGAC

Antisense: GTCCATTATCAGACATTTTAGTTTGTTCGTTTAGAGA

To assess the initiation on the different viral transcripts, and enable leader–junction site unique alignment, we used RNA Nanopore sequencing data obtained from previously SARS-CoV-2 infected (20A clade) Vero E6 monkey kidney cells [5]. This sequence was further aligned with all of the SARS-CoV-2 variants identified to date, and was confirmed by Sanger sequence analysis from a swab sample from a COVID-19-affected patient. These analyses allowed the determination of the best match sequence junction between the leader, TRS-B and the subgenomic N transcript (sgN). The 2’-*O*-methyl RNA oligonucleotides were synthetized at the DNA lab Facility at CEINGE, Biotecnologie Avanzate (Naples).

#### 4.3.2. 2’-O-Methyl RNA Oligos Targeting sgN Transfection in HEK293T-ACE2 Cells before Infection with SARS-CoV-2

HEK293T-ACE2 cells were transfected with 1 μg of sense or antisense 2’-*O*-methyl RNA oligonucleotides against SgN. Transient transfections were performed with X-tremeGENE 9 DNA Transfection Reagent (06365779001; Sigma-Aldrich). To this end, X-tremeGENE 9 DNA Transfection Reagent was equilibrated at room temperature and diluted with serum-free Dulbecco’s modified Eagle’s medium (41966-029; Gibco) (3 μL reagent/100 μL medium). Then, 1 μg of sense or antisense 2′-*O*-methyl RNA oligonucleotides were added to 100 μL of diluted X-tremeGENE 9 DNA Transfection Reagent (3:1 ratio [μL]). The transfection reagent:RNA complex was incubated for 15 min at room temperature. The transfection complex was then added to the cells in a dropwise manner. Twelve hours after transfection, the cell culture medium was changed, and the cells were infected with SARS-CoV-2 particles (GISAID accession number: EPI_ISL_736997). Forty-eight hours after transfection (i.e., after 36 h of infection), the HEK293T-ACE2 cells were lysed.

### 4.4. SARS-CoV-2 Isolation and Infection

SARS-CoV-2 was isolated from a nasopharyngeal swab obtained from an Italian patient sample as previously described [5]. Briefly, Vero E6 cells (8 × 105) were trypsinized and resuspended in Dulbecco’s modified Eagle’s medium (41966-029; Gibco) with 2% FBS in T25 flasks, to which the clinical specimen (100 μL) was added. The inoculated cultures were grown in a humidified 37 °C incubator with 5% CO_2_. Seven days after infection, when cytopathic effects were observed, the cell monolayers were scrapped with the back of a pipette tip, while the cell culture supernatant containing the viral particles was aliquoted and frozen at −80 °C. Viral lysates were used for total nucleic acid extraction for confirmatory testing and sequencing (GISAID accession number: EPI_ISL_736997).

HEK-293T cell clones stably overexpressing ACE2 (8 × 10^5^ cells) were plated in T25 flasks for transfection with sense or antisense 2’-*O*-methyl RNA oligos targeting sgN. After 12 h, the cell culture medium was changed, and the 2’-*O*-methyl RNA transfected cells were then infected with viral particles of the 20I/501Y.V1 (B.1.1.7) clades (0.03 MOI), (GISAID accession number: EPI_ISL_736997). Uninfected cells were used as the negative control. After 36 h of infection, the cells were lysed or fixed. These experiments were performed in a BLS3-authorized laboratory.

### 4.5. RNA Extraction and qPCR from Oro/Nasopharyngeal Swabs from Patients

Oro/nasopharyngeal swab samples (200 μL) were taken for RNA extraction using nucleic acid extraction kits (T-1728; ref: 1000021043; MGI tech) with automated procedures on a high-throughput automated sample preparation system (MGISP-960; MGI Tech), following the manufacturer’s instructions. RNA samples (5 μL) were used to perform qPCR with the SARS-CoV-2 Viral3 kit (BioMol laboratories) and the IVD-approved Allplex 2019-nCoV assay (Seegene).

Allplex 2019-nCoV assay for viral E, RdRP, and N gene detection. This diagnostic kit provides a specific quantitative detection of the viral E, RdRP, and N gene (from the SARS-CoV-2) by using differentially labeled target probes into two mixes (E, FAM; RdRp, Cal Red 610; N: Quasar 670) N2, VIC; Second mix: N3, VIC; RNase P, CY5). These runs were performed by using 5 μL RNA on a PCR machine (CFX96; BioRad; in vitro diagnostics IVD approved) under the following conditions:○Reverse transcription: 50 °C for 20 min;○Denaturation: 95 °C for 15 min;○Denaturation and annealing (×44 cycles): [95 °C for 10 s; 60 °C for 15 s, and 72 °C for 10 s].

SARS-CoV-2 Viral3 kit for viral sgN, gene E, gene Orf1ab and human RNAse P detection. These runs were performed using 5 μL RNA on a PCR machine (CFX96; BioRad; in vitro diagnostics IVD approved) under the following conditions:○UNG incubation: 25 °C for 2 min;○Reverse transcription: 50 °C for 15 min;○Inactivation/denaturation: 95 °C for 3 min;○Denaturation and annealing (for 44 cycles): [95 °C for 3 s and 60 °C for 45 s].

SARS-CoV-2 Viral3 kit was produced by following CDC 2019-Novel Coronavirus Real-Time RT-PCR Diagnostic Panel and WHO-technical-guidance for oligo sequences (Sequences MT810943.1. and NM_001104546.2). The details of the primers used in these assays (SARS-CoV-2 Viral3 kit) are provided below:sgN Forward: CAACCAACTTTCGATCTCTTGTAsgN Reverse: TCTGCTCCCTTCTGCGTAGAsgN Probe: 5′-FAM-ACTTCCTCAAGGAACAACATTGCCA-BBQ1-3′Orf1ab Forward: CCCTGTGGGTTTTACACTTAAOrf1ab Reverse: ACGATTGTGCATCAGCTGAOrf1ab Probe:5’-ROX- CCGTCTGCGGTATGTGGAAAGGTTATGG-BBQ2-3’E Forward: ACAGGTACGTTAATAGTTAATAGCGTE Reverse: ATATTGCAGCAGTACGCACACAE Probe: 5’ CY5-ACACTAGCCATCCTTACTGCGCTTCG BBQ2-3’RNAse P Forward: ATGGCGGTGTTTGCAGATTTRNAse P Reverse: AGCAACAACTGAATAGCCAAGGRNAse P Probe: 5′-HEX-TTCTGACCTGAAGGCTCTGCGCG-BHQ1-3′

Taqman assays for viral sgE and human β-Actin (ACTB) detection. Reverse transcription was performed with SuperScript IV VILO Master Mix (11756500, Invitrogen, Carlsbad, CA, USA), following the manufacturer’s instructions. The reverse transcription products (cDNA) were amplified by qRT-PCR using an RT-PCR system (7900; Applied Biosystems, Foster City, CA, USA). The cDNA preparation was through the cycling method by incubating the complete reaction mix as follows:○cDNA reactions: [25 °C for 5 min and 42 °C for 30 min]○Heat-inactivation: 85 °C for 5 min○Hold stage: 4 °C

The target sgE and ACTB were detected with Taqman approach [9,11]. These runs were performed on a PCR machine (Quantstudio 12K Flex, Applied Biosystems, Waltham, MA, USA) with the following thermal protocol:○Denaturation Step: 95 °C for 20 s;○Denaturation and annealing (×50 cycles): [95 °C for 1 s and 60 °C for 20 s].

The details of the primers used in these Taqman assays are provided below:sgE Forward (Taqman) [9,11]: CGATCTCTTGTAGATCTGTTCTCsgE Reverse (Taqman) [9,11]: ATATTGCAGCAGTACGCACACAsgE probe (Taqman) [9,11]: 5′-FAM-ACACTAGCCATCCTTACTGCGCTTCG-BBQ-3′

ACTB Forward (Taqman): Hs01060665_g1 (Thermo Fisher Scientific, Waltham, Massachusetts, USA)

The quantification cycle (Cq) values of sgN and sgE are reported as means ± SD normalized to the control (human ACTB) of three replicates.

### 4.6. RNA Extraction and qPCR from HEK-293T Cells Overexpressing ACE2

RNA samples were extracted with TRIzol RNA isolation reagent according to the manufacturer’s instructions. Reverse transcription was performed with 5× All-In-One RT MasterMix (catalog no. g486; ABM), according to the manufacturer’s instructions. The reverse transcription products (cDNA) were amplified by qRT-PCR using an RT-PCR system (7900; Applied Biosystems, Foster City, CA, USA). The cDNA preparation was through the cycling method by incubating the complete reaction mix as follows:○cDNA reactions: [25 °C for 5 min and 42 °C for 30 min];○Heat-inactivation: 85 °C for 5 min;○Hold stage: 4 °C.

Viral N and human RNAse P detection. The detection of viral N gene and human RNase P was performed using the in vitro diagnostics IVD-approved “Quanty COVID-19” kit (RT-25; Clonit). This diagnostic kit provides a specific quantitative detection of the viral N1, N2, and N3 fragments (from the SARS-CoV-2 N gene) and human RNaseP gene by using differentially labeled target probes into two mixes (First mix: N1, FAM; N2, VIC; Second mix: N3, VIC; RNase P, CY5). These runs were performed on a PCR machine (CFX96; Bio-Rad; IVD approved) according to the manufacturer’s instructions. Briefly, 5 μL RNA was used for the following thermal protocol below:○UNG incubation: 25 °C for 2 min;○Reverse transcription: 50 °C for 15 min;○Inactivation/denaturation: 95 °C for 2 min;○Denaturation and annealing (×45 cycles): [95 °C for 3 s and 55 °C for 30 s].

The quantification cycle (Cq) values of N1, N2, and N3 are reported as means ±SD normalized to the internal control (human RNase P) of three replicates.

Viral E, human ACE2, and β-Actin (ACTB) detection (SYBR green). The targets E, ACE2, and ACTB were detected with SYBR green approach by using BrightGreen 2X qRT-PCR MasterMix Low-ROX (MasterMix-LR; ABM). Human ACTB was used as the housekeeping gene used to normalize the quantification cycle (Cq) values of the other genes. These runs were performed on a PCR machine (Quantstudio5, Lifetechnologies) with the following thermal protocol:○Hold stage: 50 °C for 2 min;○Denaturation step: 95 °C for 10 min;○Denaturation and annealing (×45 cycles): [95 °C for 15 s and 60 °C for 60 s];○Melt curve stage: [95 °C for 15 s, 60 °C for 1 min, and 95 °C for 15 s].

The details of the primers used in these SYBR green assays are provided below:E Forward (SYBR green) [5]: ACAGGTACGTTAATAGTTAATAGCGTE Reverse (SYBR green) [5]: ATATTGCAGCAGTACGCACACAACE2 Forward (SYBR green) [5]: GAAATTCCCAAAGACCAGTGGAACE2 Reverse (SYBR green) [5]: CCCCAACTATCTCTCGCTTCATACTB Forward (SYBR green) [5]: GACCCAGATCATGTTTGAGACCTTACTB Reverse (SYBR green) [5]: CCAGAGGCGTACAGGGATAGC

The relative expression of the target genes was determined using the 2^−ΔΔCq^ method, as the fold increase compared with the controls. The data are presented as means ± SD of the 2^−ΔΔCq^ values (normalized to human ACTB) of three replicates.

*Viral sgN, sgE and human β-Actin (ACTB) detection (Taqman).* The target sgN, sgE, and ACTB were detected with the Taqman approach [5]. These runs were performed on a PCR machine (Quantstudio 12K Flex, Appliedbiosystems) with the following thermal protocol:○Denaturation Step: 95 °C for 20 s;○Denaturation and Annealing (×50 cycles): [95 °C for 1 s and 60 °C for 20 s].

The details of the primers used in these Taqman assays are provided below:sgN Forward (Taqman) [11]: CAACCAACTTTCGATCTCTTGTAsgN Reverse (Taqman) [11]: TCTGCTCCCTTCTGCGTAGAsgN Probe (Taqman) [11]: 5′FAM-ACTTCCTCAAGGAACAACATTGCCA-BBQ-3′sgE Forward (Taqman) [9,11]: CGATCTCTTGTAGATCTGTTCTCsgE Reverse (Taqman) [9,11]: ATATTGCAGCAGTACGCACACAsgE probe (Taqman) [9,11]: 5′-FAM-ACACTAGCCATCCTTACTGCGCTTCG-BBQ-3′

ACTB Forward (Taqman): Hs01060665_g1 (Thermo Fisher Scientific)

The quantification cycle (Cq) values of sgN and sgE are reported as means ± SD normalized to the control (human ACTB) of three replicates.

### 4.7. Sanger Sequencing

The cDNA was obtained by random primer RT-PCR using SensiFASTcDNA synthesis kits (Bioline, provided by Life Technologies Italia, Monza, MB, Italy), using 5 μL RNA extracted from nasopharyngeal swabs. We used the primer setting (sgN-For AAAC- CAACCAACTTTCGATCTCTTGTA and sgN-Rev TCTGGTTACTGCCAGTTGAATC) to amplify the sgN region, and to perform the Sanger sequencing.

### 4.8. Immunoblotting

Cells were lysed in 20 mM sodium phosphate, pH 7.4, 150 mM NaCl, 10% (*v*/*v*) glycerol, 1% (*w*/*v*) sodium deoxycholate, 1% (*v*/*v*) Triton X-100, supplemented with protease inhibitors (Roche). The cell lysates were cleared by centrifugation at 16,200× *g* for 30 min at room temperature, and the supernatants were removed and assayed for protein concentrations with protein assay dye reagent (Bio-Rad Laboratories, Berkeley, CA, USA). The cell lysates (20 μg) were resolved on 10% SDS-PAGE gels. The proteins were transferred to PVDF membranes (Millipore). After 1 h in blocking solution with 5% (*w*/*v*) dry milk fat in Tris-buffered saline containing 0.02% [*v*/*v*] Tween-20, the PVDF membranes were incubated with the primary antibody overnight at 4 °C: anti-ACE2 (1:1000; ab15348), anti-SARS-CoV-2 N protein (1:250; 35-579; ProSci Inc., Poway, San Diego, CA, USA), or anti-β-actin (1:10,000; A5441; Sigma-Aldrich, St. Louis, MO, USA). The membranes were then incubated with the required secondary antibodies for 1 h at room temperature: secondary mouse or rabbit horseradish-peroxidase-conjugated antibodies (NC 15 27606; ImmunoReagents, Inc.), diluted in 5% (*w*/*v*) milk in TBS-Tween. The protein bands were visualized by chemiluminescence detection (Pierce-Thermo Fisher Scientific Inc., Rockford, IL, USA). Densitometry analysis was performed with the ImageJ software. The peak areas of the bands were measured on the densitometry plots, and the relative proportions (%) were calculated. Then, the density areas of the peaks were normalized with those of the loading controls, and the ratios for the corresponding controls are presented as fold-changes. Immunoblotting was performed in triplicate. The densitometry analyses shown were derived from three independent experiments.

### 4.9. Immunofluorescence

SARS-CoV-2-infected HEK293T-ACE2 cells were fixed in 4% paraformaldehyde in phosphate-buffered saline (PBS) for 30 min, washed three times with PBS, and permeabilized for 15 min with 0.1% Triton X-100 (215680010; Acros Organics, Thermo Fisher Scientific, Waltham, MA, USA) diluted in PBS. The cells were then blocked with 3% bovine serum albumin (A9418; Sigma-Aldrich, St. Louis, MO, USA) in PBS for 1 h at room temperature. The samples were incubated with the appropriate primary antibodies overnight at 4 °C: anti-ACE2 (1:1000; ab15348; Abcam. Cambridge, UK) or anti-SARS S protein (1:100; ab272420; Abcam). After washing with PBS, the samples were incubated with the secondary antibody at room temperature for 1 h: anti-mouse Alexa Fluor 488 (1:200; ab150113; Abcam) or anti-rabbit Alexa Fluor 647 (1:200; ab150075; Abcam). DNA was stained with DAPI (1:1000; #62254; Thermo Fisher). The slides were washed and mounted with cover slips with 50% glycerol (G5150; Sigma-Aldrich). Microscopy images were obtained using the Elyra 7 platform (Zeiss) with the optical Lattice SIM^2^ technology (with the ZEN software, Zeiss, blue edition), using the 63× oil immersion objective.

### 4.10. Statistical Analysis

Statistical significance was defined as *p* < 0.05 by unpaired two-tailed student’s *t*-tests. All of the data are given as means ± SD. In the Figures, statistical significance is represented as follows: * *p* < 0.05, ** *p* < 0.01, and *** *p* < 0.001.

For the determination of the limit of detection (cut-off) for sgN detection, one-way analysis of variance (ANOVA) was used through IBM SPSS Statistics. Briefly, the samples were stratified into three groups according to the Cq values of sgN. The first group consisted of those samples where the Cq t value for sgN was below the median value (i.e., 30.51; 99 samples). The second group of samples were characterized by Cq values of sgN ranging from the Cq median value (30.51) to 40 (96 samples). The third group comprised the samples in which sgN was not detectable (i.e., Cq > 40; 120 samples). All the experiments were performed in triplicate.

## 5. Conclusions

In conclusion, we demonstrate here the importance of sgN as a new “marker” during the follow-up of hospitalized and home-isolated COVID-19-positive patients, to monitor their disease progression and therapeutic responses, which can be further correlated with their level of transmission. Furthermore, we also show in vitro the antiviral effectiveness of targeting sgN via a 2′-*O*-methyl RNA oligonucleotide against the B.1.1.7 VOC Alpha variant, which thus represents a novel therapeutic strategy against SARS-CoV-2 biogenesis that may facilitate antiviral vaccine development and drug design.

Our data provide a possible way to use gene therapy for all SARS-type viruses and to address any therapeutics in newly occurring viral infections and with the future evolution of the COVID-19 pandemic.

## 6. Patents

PCT/EP2021/087512 deposited 23-12-2021: “Method for Determining Active SARS-CoV-2 Infections”.

European Patent/EP22154018 deposited 28-1-2022: “Antisense Compounds for the Treatment Of Coronavirus Infection”.

## Figures and Tables

**Figure 1 ijms-23-01941-f001:**
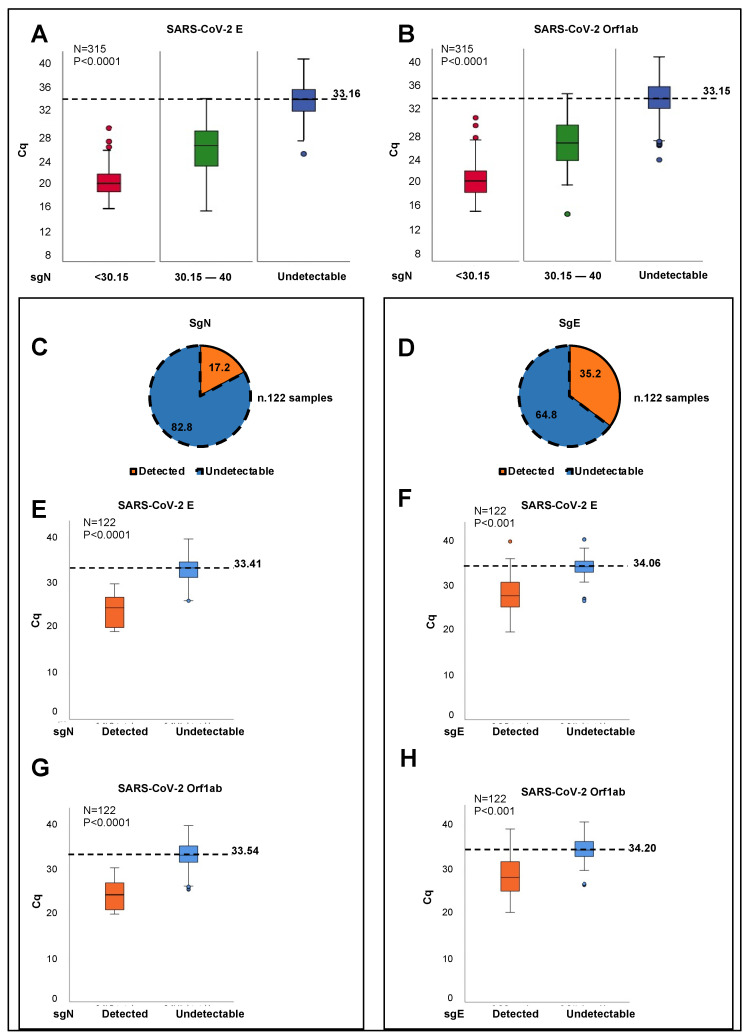
SgN and sgE are detected in samples from COVID-19-affected patients with high SARS-CoV-2 viral load. (**A**,**B**) Samples obtained from oro/nasopharyngeal swabs from COVID-19-positive patients (N = 315) were stratified into three groups according to the median Cq values of sgN (sgN Cq median = 33.51), as detected through SARS-CoV-2 Viral3 kit. The first group consisted of those samples where Cq for sgN was below the median value (i.e., Cq < 30.51; N = 99 samples; red). The second group was characterized by Cq values for sgN from the Cq median value (30.51) to 40.00 (N = 96 samples; green). The third group comprised samples where sgN was not detected (i.e., Cq > 40; *N* = 120 samples; blue). An ANOVA was used through IBM SPSS Statistics to determine the cut-off for sgN detection. SgN was detected in the samples with viral E Cq values < 33.163 (**A**) and ORF1ab Cq values < 33.155 (**B**) (*p* < 0.0001, for both). (**C**,**D**) Pie charts showing the proportions (%) of the oro/nasopharyngeal swab specimens where the levels of sgN (**C**) and sgE (**D**) were detectable (i.e., Cq values < 40; orange) or not detectable (i.e., Cq values > 40; blue), for the 122 COVID-19-positive patients belonging to a single cohort (entire cohort, N = 315). The data show no detectable levels of sgN and sgE in 82.8% (**C**; blue) and 64.8% (**D**; blue) of the patients, respectively. (**E**–**H**) An ANOVA was used through IBM SPSS Statistics to determine the cut-off for sgN and sgE detection in the 122 oro/nasopharyngeal swabs from the single-cohort COVID-19-positive patients. SgN was detected in the samples with viral E Cq values < 33.41 (**E**) and ORF1ab Cq values <33.54 (**G**) (*p* < 0.0001, for both). SgE was detected in the samples with viral E Cq values <34.06 (**E**) and ORF1ab Cq values < 34.20 (**G**) (*p* < 0.001, for both).

**Figure 2 ijms-23-01941-f002:**
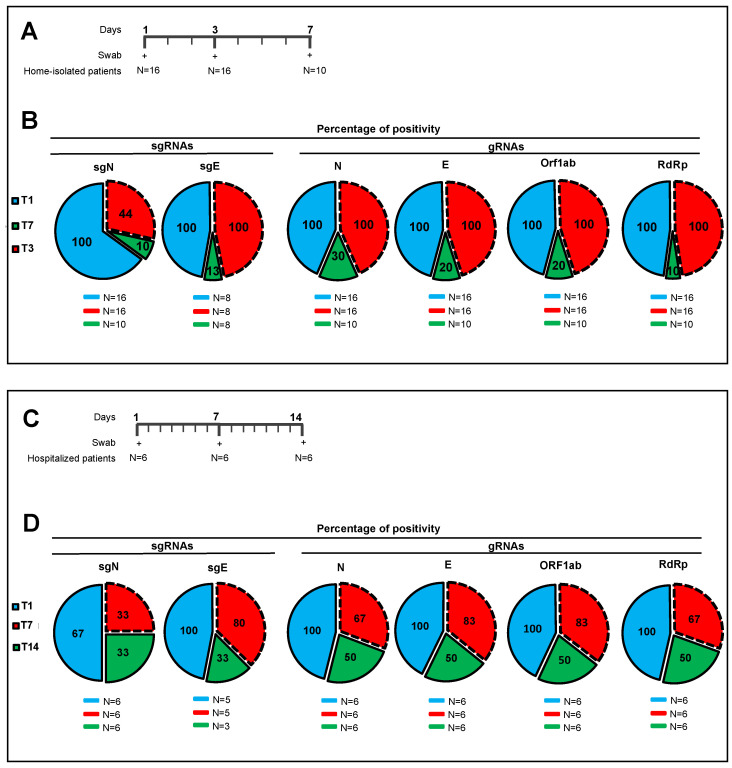
Loss of detection of sgN precedes SARS-CoV-2 replication failure in home-isolated and hospitalized COVID-19-affected patients. (**A**) A cohort of oro/nasopharyngeal swabs was collected from home-isolated COVID-19-positive patients and analyzed according to the scheduled times (i.e., at 3-day intervals from the first swab). Ten patients were followed up to 7 days from the first swab test; 6 patients were followed up to 3 days. (**B**) Pie charts showing the proportions (%) of positivity of the oro/nasopharyngeal samples to viral subgenomic sgN and sgE, and genomic N, E, ORF1ab, and RdRp at the different times (blue, first swab [N = 16]; red, second swab collected after 3 days [N = 16]; green, third swab collected after 7 days [N = 10]). SgE was detected in 8 oro/nasopharyngeal samples. SgN was detected in 44% of the samples after 3 days from the first swab. SgN, gene E, and gene ORF1ab were measured using the SARS-CoV-2 Viral3 kit. Gene N, gene E, and gene ORF1ab were detected using the Allplex 2019-nCoV assay. SgE was evaluated by Taqman qPCR. (**C**) A cohort of oro/nasopharyngeal swabs collected from 6 hospitalized COVID-19-positive patients was analyzed according to the scheduled times (i.e., 7-day intervals from the first swab). (**D**) Pie charts showing the proportions (%) of positivity of the oro/nasopharyngeal samples to viral subgenomic sgN and sgE, and genomic N, E, ORF1ab, and RdRp at the different time points (blue, first swab [N = 6]; red, second swab collected after 7 days [N = 6]; green, third swabs collected after 14 days [N = 6]). SgE was detected in 5 oro/nasopharyngeal samples. SgN was detected in 50% of the samples after 7 days from the first swab. sgN, gene E, and gene ORF1ab were measured using the SARS-CoV-2 Viral3 kit. N and ORF1ab were detected using the Allplex 2019-nCoV assay. SgE was evaluated by Taqman qPCR.

**Figure 3 ijms-23-01941-f003:**
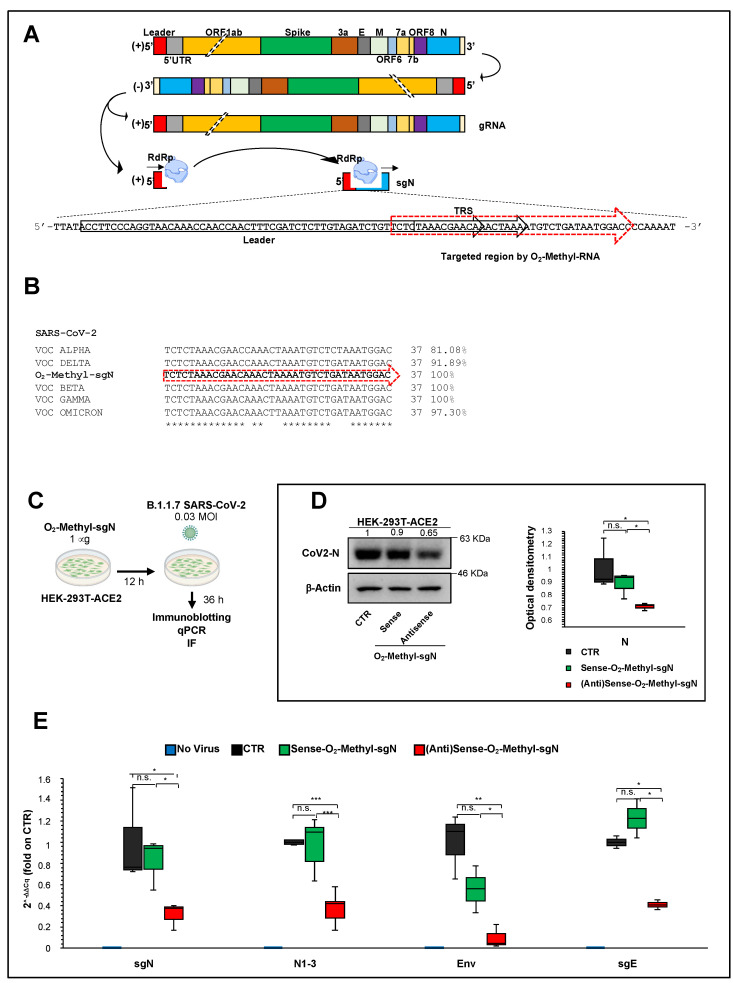
Reduction of viral load by targeting sgN in HEK293T-ACE2 SARS-CoV-2-infected cells: (**A**) Schematic representation of the SARS-CoV-2 genome organization, the canonical sgRNAs biogenesis (according to the “leader-to-body fusion” model) and the 2’-O-methyl antisense RNA design. During its viral cycle, SARS-CoV-2 virus replicates its positive sense (+) genomic RNA (29,903 nt) to produce full-length negative-sense (−) RNA molecules that act as templates for the synthesis of new positive-sense (+) gRNAs, which are then packaged into newly assembled virions. However, according to the “leader-to-body fusion” model [1], during negative-strand synthesis, the replication complex (including RdRp) interrupts the transcription when it crosses a transcription-regulatory sequence (TRS) located upstream to most ORFs (including gene N) in the 3′ one-third of the viral genome (TRS ‘body’, or TRS-B). Thus, the synthesis of the negative-strand (−) RNA is re-initiated at the TRS in the leader sequence (TRS-L, red box) at 70 nucleotides from the 5′ end of the genome. Through this mechanism, a negative strand copy of the leader sequence is added to the nascent RNA to complete the synthesis of negative-strand (−) sgRNAs. These negative RNAs are then used as templates to synthesize positive-sense (+) sgRNAs that are translated into both structural and accessory proteins. 2’-*O*-Methyl antisense RNA (37 bp, yellow arrow with dashed lines) was developed based on the sequence of sgN (i.e., leader sequence [in red], TRS upstream to gene N [in green], and 5′ end of gene N). Adapted from “Viral genome (SARS-CoV-2), by BioRender (2021); Retrieved from https://app-biorender.com/biorender-templates (accessed on 30 December 2021). (**B**) Sequence alignment of the regions recognized by 2′-*O*-methyl RNA sgN among SARS-CoV-2 VOC Alpha, Beta, Gamma, Delta, and Omicron, illustrating their degrees of percentage identity (i.e., VOC Alpha: 81.1%; Beta: 100%; Gamma: 100%; Delta: 91.9%; Omicron: 97.3%). These sequence alignments were realized using the Clustal Omega software. (**C**) Experimental plan. Human HEK-293T cells overexpressing ACE2 were plated and transfected with 2’-*O*-methyl antisense or sense RNA against sgN. Nontransfected cells were used as the negative control. After 12 h, the cells were infected with 20I/501Y.V1 (B.1.1.7) (UK) (EPI_ISL_736997) SARS-CoV-2 viral particles (MOI, 0.03), and noninfected cells were used as the negative control for infection. At 36 h from infection (i.e., 48 h from transfection), the cells were lysed or fixed (in 4% paraformaldehyde) for protein/RNA extraction or immunofluorescence, respectively. (**D**) Left: Representative immunoblot (using antibodies against the indicated proteins) of human HEK-293T cells overexpressing ACE2 transfected with 2’-*O*-methyl antisense or sense RNA against sgN and then infected with 20I/501Y.V1 (B.1.1.7) (UK) (EPI_ISL_736997) SARS-CoV-2 viral particles (MOI, 0.03) for a further 36 h. β-Actin was used as the loading control. All experiments were performed in triplicate. Right: Densitometry analysis of the indicated band intensities in blots from three independent experiments. Data are means ± SD. * *p* < 0.05 (unpaired two-tailed student’s *t* test; N = 3 independent experiments per group). Negative controls: nontransfected cells (CTR). N.S., not significant. I Quantification of mRNA abundance relative to that in control (CTR) cells (2^−ΔΔCq^) for sgN, sgE, N1-3 and gene E. RT-PCR analysis of RNA extracted from cells treated as described in (**E**). Noninfected cells and SARS-CoV-2-infected cells not transfected (CTR) were used as controls. Data are means ± SD. * *p* < 0.05, ** *p* < 0.01, and *** *p* < 0.001 (unpaired two-tailed student’s *t* test; N = 3 independent experiments per group).

**Figure 4 ijms-23-01941-f004:**
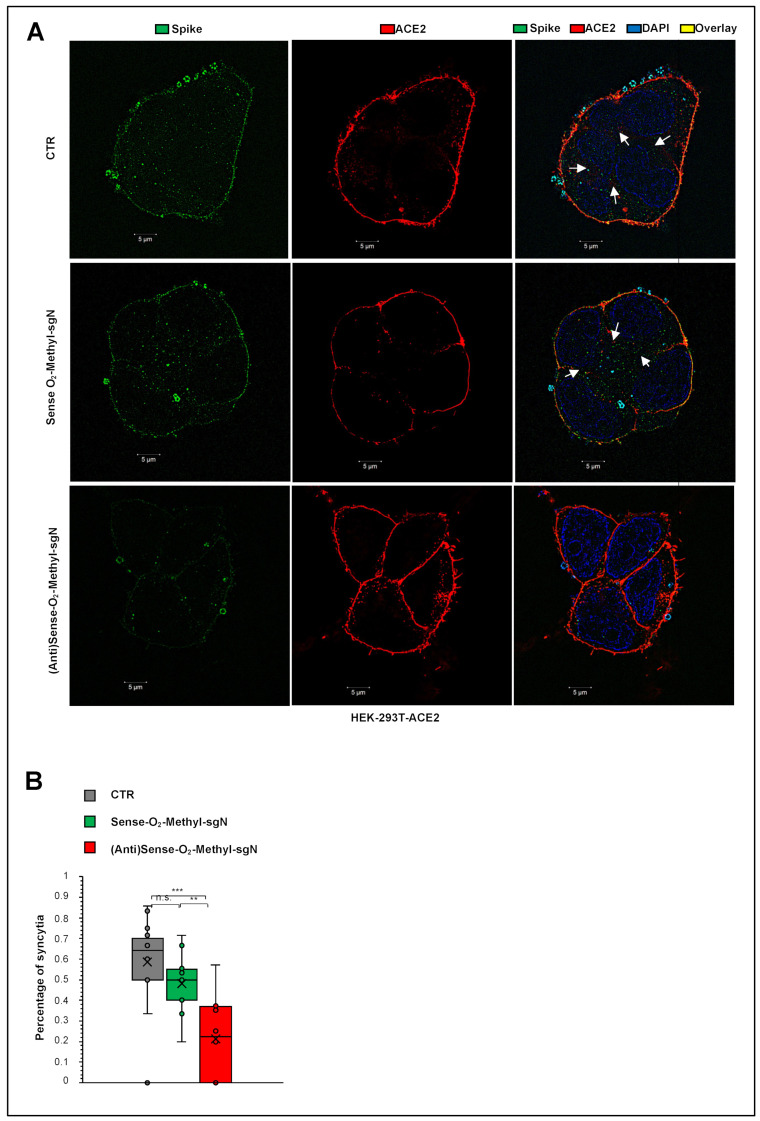
Reduction of syncytia formation by targeting sgN in HEK293T-ACE2 SARS-CoV-2-infected cells. (**A**) Immunofluorescence staining with an antibody against the viral S protein (green) and human ACE2 protein (red) in human HEK-293T cells overexpressing ACE2 transfected with 2’-*O*-methyl antisense or sense RNA against sgN for 12 h and then infected with 20I/501Y.V1 (B.1.1.7) (UK) (EPI_ISL_736997) SARS-CoV-2 viral particles (MOI, 0.03) for a further 36 h. SARS-CoV-2-infected cells not transfected (CTR) were used as the negative control. Representative images of syncytia. DAPI was used for nuclei. The SIM images were acquired with Elyra 7 and processed with the Zeiss ZEN software (blue edition). Magnification, ×63. Scale bars: 5 μm. (**B**) Comparison of the proportions (%) of syncytia counted by SIM fluorescence microscopy in human ACE2 (red) proteins in human HEK-293T cells overexpressing ACE2 transfected with 2′-*O*-methyl antisense or sense RNA against sgN for 12 h and then infected with 20I/501Y.V1 (B.1.1.7) (UK) (EPI_ISL_736997) SARS-CoV-2 viral particles (MOI, 0.03) for a further 36 h. SARS-CoV-2-infected cells not transfected (CTR) were used as the negative control. Data are means ± SD. ** *p* < 0.01, and *** *p* < 0.001 (unpaired two-tailed student’s *t* test; N = 3 independent experiments per group). More than 60 nuclei were counted per condition.

**Figure 5 ijms-23-01941-f005:**
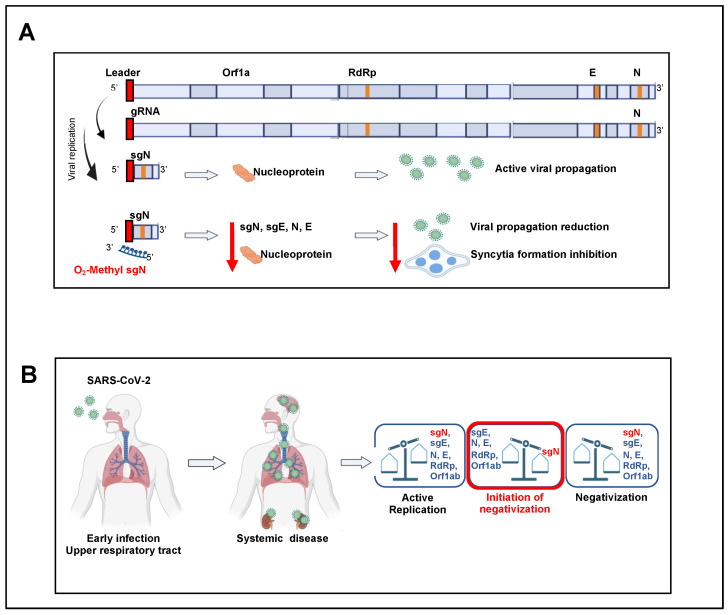
Proposed mechanism of action of sgN during infection by SARS-CoV-2. (**A**) The biogenesis of subgenomic RNAs follows the “leader to TRS-B fusion model”, which produces copies of sgN that are translated to the N protein. Impairment of this process through 2’-*O*-methyl antisense RNA against sgN results in therapeutic benefits in terms of decreased viral replication and syncytia formation in vitro. (**B**) During SARS-CoV-2 infection, the acute phase of virus replication is enhanced by expression of viral genome and subgenomic transcripts; while during the negativization process, the loss of sgN detection is seen (i.e., undetectable levels: Cq > 40; viral ORF1ab and E Cq values > 33.15 and 33.16, respectively).

## Data Availability

These sequences have been deposited at Sequence Read Archive SRA–Bio Project PRJNA688696.

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
