# Peer review of "Loss of Detection of sgN Precedes Viral Abridged Replication in COVID-19-Affected Patients—A Target for SARS-CoV-2 Propagation"

_ijms, 2022, doi:10.3390/ijms23041941_

Round 1

Reviewer 1 Report

The authors reported about the significant decrease of sgN level in the early stage of infection. The detection methods that the authors established may be useful for assessing whether the patients have potential to spread the SARS-CoV-2. Also, the authors reported about possible therapy using 2'-O-methyl antisense RNA. This manuscript is well written and providing valuable information on SARS-CoV-2 infection and its therapy. There are minor comments from the reviewer.

-Please mention in materials and methods that how the original number of viral copies or PFU were counted in this study.

-Is there any information about sensitivity of the authors detection methods against various SARS-CoV-2 strains?

Figure 2B, 2D

The pie figures are not seem to be suitable for these data. The authors should change them to another type of figure, such as bar graphs or Line graphs.

Figure 3E

Fix Y axis “2^-DDCt” to “2-ΔΔCT”

Author Response

Response to Reviewer 1 Comments

Point 1: Ln246-247:  The statement that loss of detection of sgN is predictive of viral negativization is not directly shown by this data.  The data shows that it is the first target lost in the testing, but this could be due to multiple reasons such as loss of stability, less efficient RT-PCR.  I would suggest rewording this statement and discuss other possible outcomes as to why sgN is lost initially in these patients.

Response 1: We thank reviewer for his observations which will clarify the manuscript findings. We have now rephrased the statement in the Results section as follow (Ln247-260):

Several reasons can be ascribed to the loss of sgN detection, and the most appropriate could be due to its rapid degradion if compared to other viral markers. However, litterature data show that sgRNAs are protected thought “double-membrane vesicles” from the hydrolytic actions of intracellular nucleases (see Alexandersen at al. [6]). Moreover, due to the specificity (≥ 99.9%) and sensitivity (1.5 copies/ml for Allplex 2019-nCoV and 30 viral copies/ml for SARS-CoV-2 Viral3, see Supplementary figures S1B) of the commercial kit here used, its unlikely that only sgN has a higher rate of degradation compared to the other markers (Allplex 2019-nCoV: gene N, gene E , gene RpRd; SARS-CoV-2 Viral3: sgN, gene E, gene ORF1ab). Furthermore, we have previously reported a direct correlation between sgN expression and the viral load (MOI) in SARS-CoV-2-infected Vero E6 cells at different viral particle numbers [11]. However, at this time, we cannot exclude an additional hypothesis of a potential higher rate of instability and less efficient RT-PCR detection for sgN RNA in comparison to the others viral genomic and subgenomic targets”.

Furthermore, we have now discussed other possible outcomes as to why sgN is lost initially in these patients in the Discussion section (Ln465-479):

“The N protein, which is the only protein present in the coronavirus nucleocapsid, plays a critical role in ensuring coronavirus replication and successful intracellular lifecycle, thus it is considered a suitable target designing new vaccines together with S-RBD protein [22-23]. At this time, a question can be raised on why do we observe a loss of sgN before the other markers during the time of clinical virus negativization? We thus think that this is due to the importance of sgN on supplying N protein for generating intact new genome RNA copies of the viral particles once assembled, hence underlying its functional importance on substaining the viral stability and potency. In this case, missing N protein, as observed on measuring sgN RNA copies loss, supports the hypothesis that virus replication and infection capability is diminushing as sign of negativization. At this time we cannot exclude that sgN RNA, although present in a larger copy number during viral RNA genome replication (see data presented by Nanopore sequencing, BioProject PRJNA688696, [5]) would be more sensible to RNA instability in comparison with other target viral mRNA genes. Future laboratory settings will address these hypotheses”.

Point 2:  What are the CT values for the other targets between the ones missing sgN and those detected.  Does this match the first batch of data from Figure 1?  Also, what was the sgE detection rate in hospitalized patients. 

Response 2: We thank reviewer for asking to clarify the data here presented. Regarding the first question, the CT values (now renamed as “Cq values” as requested by Reviewer 2) match the first batch of data from Figure 1 in 81.8% of the cases analyzed, as shown in Supplementary Tables 5 – 6 and Supplementary Figures S2-3. We have now added this information in the Results section (Ln293-297), as follow:

“Altogether these data show that the detection of sgN in these two independent COVID19 patients cohorts match with the viral genes E and Orf1ab cut-off for the limit of detection (showed in Figures 1A-B) in 81.8% of the cases analyzed (Supplementary Tables 5 – 6, Supplementary Figures S2-3).”

Regarding the second question “what was the sgE detection rate in hospitalized patients ?” 

Response 2.1: The expression of sgE is now available in Figure 2D, Suppementary Figure S3 and Supplementary Table 6.

The expression levels of sgE has a similar trend to the other viral genomic markers as discussed in the Results section (Ln282-289), as follow:

“We then analyzed an independent cohort of six COVID19-affected patients hospitalized in an Intensive Care Unit. Here, the analysis monitored sgN, sgE, gene E, gene ORF1ab, gene N, and RdRp using longitudinal detection at 7-day intervals (0, 7, 14 days; Supplementary Table 6). SgN was detected on the first swab tests, and again in 67% of the patients after 7 days, and in 33% after 14 days (Figure 2C, D, Supplementary Figure S3A, B). Then there was loss of detection of the other viral genes in these patients after 7 days and 14 days (detected in, respectively: sgE 80%, 33%; gene E, 83%, 50%; gene ORF1ab, 83%, 50%; gene N, 67%, 50%; RdRp, 67%, 50%).”

Reviewer 2 Report

Major Comments:

Ln246-247:  The statement that loss of detection of sgN is predictive of viral negativization is not directly shown by this data.  The data shows that it is the first target lost in the testing, but this could be due to multiple reasons such as loss of stability, less efficient RT-PCR.  I would suggest rewording this statement and discuss other possible outcomes as to why sgN is lost initially in these patients.

Figure 2: What are the CT values for the other targets between the ones missing sgN and those detected.  Does this match the first batch of data from Figure 1?  Also, what was the sgE detection rate in hospitalized patients. 

Author Response

Response to Reviewer 2 Comments

Point 1: I find inconsistencies in the use of PCR terminology. For example, the qPCR threshold cycle is mentioned as CT, Ct, or Cq at different regions. Better to be consistent by using Cq, which is now widely accepted.

Response 1: We thank the reviewer for his comment related the improvements of our manuscript. Following his suggestion, we have substituted the threshold cycle terminology “CT” with the quantification cycle “Cq” throughout the manuscript, the Supplemenal Information, the Figures and the Tables, including their legends.

Point 2: In bioinformatics terms, homology means origin from a common ancestor. Thus, a gene can be either homologous or not homologous (an analogy: a woman either pregnant or not pregnant). Thus, 80% and 50% homology as mentioned (e.g., Line 52, Line 360) is incorrect and requires revisions.

Response 2: We thank the reviewer for this comment and we apologize for the misinterpretation. We have now substituted “identity homology” with “percentage identity” throughout the manuscript when referred to the quantitative measurement of the similarity between two or more sequences.

Point 3: The respiratory system is primarily affected by COVID-19. What is the rationale for selecting HET-293T cells derived from the human embryonic kidney for the transfection study?

Response 3: We thank the reviewer for this comment. We have now added the rationale beyond the selection of HET-293T for SARS-CoV-2 study in the Results section (Ln301-312), as follow:

“SARS-CoV-2 has been now reported to induce systemic perturbations by impacting multiple organs. SARS-CoV-2 has been now reported to induce systemic perturbations by impacting multiple organs. In this regard, lung is the primary target for SARS-CoV-2 that causes an early respiratory infection. Then, mostly due to the wide expression of ACE2 in a heterogeneous population of systemic cells, SARS-CoV-2 can damage several systemic tissues and result in multi-organ dysfunction, including kidney due to high ACE2 expression levels [15]. HEK-293T cells have been widely used as a cellular model to identify seral mechanisms of action. Here, in order to allow and normalize SARS-CoV-2 infection, we have generated HEK-293T stable cell clones overexpressing human ACE2 cDNA under the control of CMV promoter (Supplementary Figure S4A-C). This cellular model overcomes those alterations in the viral infection efficiency due to the different multiplicity of infection (MOI). Furthermore, at this time, kidney cells (including HEK-293 expressing ACE2) are used for in vitro model platforms for SARS-CoV-2 infection [16-17].

Furthermore, we have now better described how these clones were generated and used for this study in Material and method section (point 4.2: Ln570-582; point 4.3.2: Ln599-612), as follow:

“4.2. Generation of HEK293T-ACE2 stable clones. HEK-293T cells were plated in 6-well plates in 2 ml of Dulbecco’s modified Eagle’s medium (41966-029; Gibco) with 10% fetal bovine serum (10270-106; Gibco). When the culture reached ~70% confluency, they were transfected with pCEP4-myc-ACE2 plasmid (#141185, Addgene) with X-tremeGENE 9 DNA Transfection Reagent (06365779001; Sigma-Aldrich). Briefly, X-tremeGENE 9 DNA Transfection Reagent was equilibrated at room temperature (+15 to +25°C) and diluted with serum-free Dulbecco’s modified Eagle’s medium (41966-029; Gibco) to a concentration of 3 μl reagent/100 μl medium. Then, 1 μg of DNA plasmid was added to 100 μl of diluted X-tremeGENE 9 DNA Transfection Reagent (3:1 ratio [μl]). The transfection reagent: DNA complex was then incubated for 15 minutes at room temperature. Finally, the transfection complex was added to the cells in a dropwise manner. Following forty-eight hours from transfection, the cell culture medium was changed, and the cells clones were selected using 800 mg/ml hygromycin”.

“4.3.2. 2'-O-Methyl RNA oligos targeting sgN transfection in HEK293T-ACE2 cells before infection with SARS-CoV-2. HEK293T-ACE2 cells were transfected with 1 mg of sense or anti-sense 2'-O-methyl RNA oligonucleotides against SgN. Transient transfections were performed with X-tremeGENE 9 DNA Transfection Reagent (06365779001; Sigma-Aldrich). To this end, X-tremeGENE 9 DNA Transfection Reagent was equilibrated at room temperature and diluted with serum-free Dulbecco’s modified Eagle’s medium (41966-029; Gibco) (3 μl reagent/100 μl medium). Then, 1 μg of sense or anti-sense 2'-O-methyl RNA oligonucleotides were added to 100 μl of diluted X-tremeGENE 9 DNA Transfection Reagent (3:1 ratio [μl]). The transfection reagent: RNA complex was incubated for 15 minutes at room temperature. The transfection complex was then added to the cells in a dropwise manner. Twelve hours after transfection, the cell culture medium was changed, and the cells were infected with SARS-CoV-2 particles (GISAID accession number: EPI_ISL_736997). Forty-eight hours after transfection (i.e., after 36 h of infection), the HEK293T-ACE2 cells were lysed.”

Point 4: Where did you get HEK-293T cells? What are the culture conditions such as seeding density, CO2, etc? Where did you synthesize 2’-O-Methyl RNA oligos, and who designed these oligos?

Response 4: We thank the reviewer for this comment. We have now added the missing informations regarding HEK-293T cells and their culture conditions in Materials and Methods section (point 4.1: Lm562-569), as follow:

“Cell culture: HEK-293T cells (CRL-3216, ATCC), HEK-293T stable clones overexpressing human ACE2 (HEK293T-ACE2) and Vero E6 cells (C1008, ATCC) were grown in a humidified 37°C incubator with 5% CO2. The cells were cultured in feeder-free conditions using Dulbecco’s modified Eagle’s medium (41966-029; Gibco) with 10% fetal bovine serum (10270-106; Gibco), 2 mM L-glutamine (25030-024; Gibco), and 1% penicillin/streptomycin (P0781; Sigma-Aldrich), with medium changed daily. Cells were dissociated with Trypsin-EDTA solution (T4049, Sigma-Aldrich) when the culture reached ~80% confluency.”

Furthermore, we have now included informations related to 2’-O-Methyl RNA oligos design in the Material and method section (point 4.3.1, Lm584-598), as follow:

“4.3.1. 2'-O-Methyl RNA oligos design: The following 2'-O-methyl RNA oligonucleotides were designed against the junction site between the leader sequence, the transcription-regulating sequences (TRSs) and the 5′ end of gene N (leader-“junction”-TRS-“junction”-N) of SARS-CoV-2:

Sense: TCTCTAAACGAACAAACTAAAATGTCTGATAATGGAC

Antisense: GTCCATTATCAGACATTTTAGTTTGTTCGTTTAGAGA

To assess the initiation on the different viral transcripts, and enable leader–junction site unique alignment, we used RNA Nanopore sequencing data obtained from previously SARS-CoV-2 infected (20A clade) Vero E6 monkey kidney cells [5]. This sequence was further aligned with all of the SARS-CoV-2 variants identified to date, and was confirmed by Sanger sequence analysis from a swab sample from a COVID19-affected patient. These analyses allowed the determination of the best match sequence junction between the leader, TRS-B and the subgenomic N transcript (sgN). The 2'-O-methyl RNA oligonucleotides were synthetized at DNA lab Facility at CEINGE, Biotecnologie Avanzate (Naples).”

Point 5: Line 538: Please be specific about the biological sample you used to isolate COVID-19 from the patient samples? Was it a nasopharyngeal swab?

Response 5: We thank the reviewer for this comment. The informations about the biological sample and the method used for SARS-CoV-2 isolation were added in the Materials and Methods section (point 4.4, Ln614-623), as follow:

“4.4. SARS-CoV-2 isolation and infection: SARS-CoV-2 was isolated from a nasopharyngeal swab obtained from an Italian patient sample as previously described [5]. Briefly, Vero E6 cells (8 × 105) were trypsinized and resuspended in Dulbecco’s modified Eagle’s medium (41966-029; Gibco) with 2% FBS in T25 flask to which the clinical specimen (100 ml) was added. The inoculated cultures were grown in a humidified 37°C incubator with 5% CO2 . Seven days after infection, when cytopathic effects were observed, the cell monolayers were scrapped with the back of a pipette tip, while the cell culture supernatant containing the viral particles was aliquoted and frozen at −80°C. Viral lysates were used for total nucleic acid extraction for confirmatory testing and sequencing (GISAID accession number: EPI_ISL_736997)”.

Points 6-9: Lines 551-552: What qPCR program was used? It is not clear why did you present 25° C for 2 min……....etc? Did you multiplex SgN, E, and ACE2 genes in the Taqman assay? Where is the probe sequence for sgN? It is unclear why the human housekeeping gene ACTB was used as a normalizer and to normalize which gene? Why was the human RNase P gene used? Where are the primers of these genes?

Response 6-9: We thank the reviewer for asking these clarifications. The qPCR methodology, together with the oligos and probe sequences, and the normalization analyses with human housekeeping gene ACTB or RNAse P genes were completely reformatted in the the Materials and Methods section (points 4.5 - 4.6: Ln632-761), as follow:

“4.5. RNA extraction and qPCR from oro/nasopharyngeal swabs from patients

     Oro/nasopharyngeal swab samples (200 μL) were taken for RNA extraction using Nucleic Acid Extraction kits (T-1728; ref: 1000021043; MGI tech) with automated procedures on a high-throughput automated sample preparation system (MGISP-960; MGI Tech), following the manufacturer instructions. RNA samples (5 mL) were used to perform qPCR with the SARS-CoV-2 Viral3 kit (BioMol laboratories) and the IVD-approved Allplex 2019-nCoV assay (Seegene).

     Allplex 2019-nCoV assay for viral E, RdRP, and N gene detection. This diagnostic kit provides a specific quantitative detection of the viral E, RdRP, and N gene (from the SARS-CoV-2) by using differentially labeled target probes into two mixes (E, FAM; RdRp, Cal Red 610; N: Quasar 670) N2, VIC; Second mix: N3, VIC; RNase P, CY5). These runs were performed by using 5 ml RNA on a PCR machine (CFX96; BioRad; in-vitro diagnostics IVD approved) under the following conditions:

  • Reverse Transcription: 50 °C for 20 min
  • Denaturation: 95 °C for 15 min
  • Denaturation and Annealing (×44 cycles): [95 °C for 10 secec; 60 °C for 15 s, and 72 °C for 10 sec].

     SARS-CoV-2 Viral3 kit for viral sgN, gene E, gene Orf1ab and human RNAse P detection. These runs were performed by using 5 ml RNA on a PCR machine (CFX96; BioRad; in-vitro diagnostics IVD approved) under the following conditions:

  • UNG incubation: 25°C for 2 min
  • Reverse Transcription: 50°C for 15 min
  • Inactivation/Denaturation: 95° c for 3 min
  • Denaturation and Annealing (for 44 cycles): [95°C for 3 sec and 60°C for 45 sec]

SARS-CoV-2 Viral3 kit was produced by following CDC 2019-Novel Coronavirus Real-Time RT-PCR Diagnostic Panel and WHO-technical-guidance for oligo sequences (Sequences MT810943.1. and NM_001104546.2). The details of the primers used in these assays (SARS-CoV-2 Viral3 kit) are provided below:

sgN Forward: CAACCAACTTTCGATCTCTTGTA

sgN Reverse: TCTGCTCCCTTCTGCGTAGA

sgN Probe: 5’FAM-ACTTCCTCAAGGAACAACATTGCCA-BBQ-3’

Orf1ab Forward: CCCTGTGGGTTTTACACTTAA                                 

Orf1ab Reverse: ACGATTGTGCATCAGCTGA                                     

Orf1ab Probe: CCGTCTGCGGTATGTGGAAAGGTTATGG                   

E Forward: ACAGGTACGTTAATAGTTAATAGCGT               

E Reverse: ATATTGCAGCAGTACGCACACA             

E Probe: ACACTAGCCATCCTTACTGCGCTTCG                     

RNAse P Forward: AGATTTGGACCTGCGAGCG                    

RNAse P Reverse: GAGCGGCTGTCTCCACAAGT

RNAse P Probe: 5’FAM-TTCTGACCTGAAGGCTCTGCGCG-BHQ1-3’

      Taqman assays for viral sgE and human b-Actin (ACTB) detection. Reverse transcription was performed with SuperScript IV VILO Master Mix (11756500, Invitrogen, Carlsbad, CA, USA), following the manufacturer’s instructions. The reverse transcription products (cDNA) were amplified by qRT-PCR using an RT-PCR system (7900; Applied Biosystems, Foster City, CA, USA). The cDNA preparation was through the cycling method by incubating the complete reaction mix as follows:

  • cDNA reactions: [25 °C for 5 min and 42 °C for 30 min]
  • Heat-inactivation: 85°C for 5 min
  • Hold stage: 4 °C

The target sgE and ACTB were detected with Taqman approach [9, 11]. These runs were performed on a PCR machine (Quantstudio 12K Flex, Appliedbiosystems) with the following thermal protocol:

  • Denaturation Step: 95 °C for 20 sec
  • Denaturation and Annealing (×50 cycles): [95°C for 1 s and 60 °C for 20 s].

The details of the primers used in these Taqman assays are provided below:

sgE Forward (Taqman) [9, 11]: CGATCTCTTGTAGATCTGTTCTC

sgE Reverse (Taqman) [9,11]: ATATTGCAGCAGTACGCACACA

sgE probe (Taqman) [9, 11]: 5’FAM-ACACTAGCCATCCTTACTGCGCTTCG-BBQ-3’

ACTB Forward (Taqman): Hs01060665_g1 (Thermo Fisher Scientific)

The quantification cycle (Cq) values of sgN and sgE are reported as means ±SD normalized   

to the control (human ACTB) of three replicates.

4.6. RNA extraction and qPCR from HEK-293T cells overexpressing ACE2

       RNA samples were extracted with TRIzol RNA Isolation Reagent according to the manufacturer instructions. Reverse transcription was performed with 5× All-In-One RT MasterMix (catalog no. g486; ABM), according to the manufacturer instructions. The reverse transcription products (cDNA) were amplified by qRT-PCR using an RT-PCR system (7900; Applied Biosystems, Foster City, CA, USA). The cDNA preparation was through the cycling method by incubating the complete reaction mix as follows:

  • cDNA reactions: [25 °C for 5 min and 42 °C for 30 min]
  • Heat-inactivation: 85°C for 5 min
  • Hold stage: 4 °C

        Viral N and human RNAse P detection. The detection of viral N gene and human RNase P was performed by using the in-vitro diagnostics IVD-approved approved “quanty COVID19” kit (RT-25; Clonit). This diagnostic kit provides a specific quantitative detection of the viral N1, N2, and N3 fragments (from the SARS-CoV-2 N gene) and human RNaseP gene by using differentially labeled target probes into two mixes (First mix: N1, FAM; N2, VIC; Second mix: N3, VIC; RNase P, CY5). These runs were performed on a PCR machine (CFX96; Bio-Rad; IVD approved) according to manufacturer instructions. Briefly, 5 ml RNA was used for the following thermal protocol below:

  • UNG incubation: 25°C for 2 min
  • Reverse Transcription: 50°C for 15 min
  • Inactivation/ Denaturation: 95 °C for 2 min
  • Denaturation and Annealing (×45 cycles): [95°C for 3 sec and 55 °C for 30 sec].

The quantification cycle (Cq) values of N1, N2, and N3 are reported as means ±SD normalized to the internal control (human RNase P) of three replicates.

       Viral E, human ACE2 and b-Actin (ACTB) detection (SYBR green). The targets E, ACE2 and ACTB were detected with SYBR green approach by using BrightGreen 2X qRT-PCR MasterMix Low-ROX (MasterMix-LR; ABM). Human ACTB was used as the housekeeping gene used to normalize the quantification cycle (Cq) values of the other genes. These runs were performed on a PCR machine (Quantstudio5, Lifetechnologies) with the following thermal protocol:

  • Hold stage: 50°C for 2 min
  • Denaturation Step: 95 °C for 10 min
  • Denaturation and Annealing (×45 cycles): [95°C for 15 sec and 60 °C for 60 sec].
  • Melt curve stage: [95°C for 15 sec, 60°C for 1 min and 95°C for 15 sec]

The details of the primers used in these SYBR green assays are provided below:

E Forward (SYBR green) [5]: ACAGGTACGTTAATAGTTAATAGCGT

E Reverse (SYBR green) [5]: ATATTGCAGCAGTACGCACACA

ACE2 Forward (SYBR green) [5]: GAAATTCCCAAAGACCAGTGGA

ACE2 Reverse (SYBR green) [5]: CCCCAACTATCTCTCGCTTCAT

ACTB Forward (SYBR green) [5]: GACCCAGATCATGTTTGAGACCTT

ACTB Reverse (SYBR green) [5]: CCAGAGGCGTACAGGGATAGC

The relative expression of the target genes was determined using the 2−ΔΔCq method, as 

the fold increase compared with the controls. The data are presented as means ±SD of the

2−ΔΔCq values (normalized to human ACTB) of three replicates.

        Viral sgN, sgE and human b-Actin (ACTB) detection (Taqman). The target sgN, sgE and ACTB were detected with Taqman approach [5]. These runs were performed on a PCR machine (Quantstudio 12K Flex, Appliedbiosystems) with the following thermal protocol:

  • Denaturation Step: 95 °C for 20 sec
  • Denaturation and Annealing (×50 cycles): [95°C for 1 s and 60 °C for 20 s].

The details of the primers used in these Taqman assays are provided below:

sgN Forward (Taqman) [11]: CAACCAACTTTCGATCTCTTGTA

sgN Reverse (Taqman) [11]: TCTGCTCCCTTCTGCGTAGA

sgN Probe (Taqman) [11]: 5’FAM-ACTTCCTCAAGGAACAACATTGCCA-BBQ-3’

sgE Forward (Taqman) [9, 11]: CGATCTCTTGTAGATCTGTTCTC

sgE Reverse (Taqman) [9,11]: ATATTGCAGCAGTACGCACACA

sgE probe (Taqman) [9, 11]: 5’FAM-ACACTAGCCATCCTTACTGCGCTTCG-BBQ-3’

ACTB Forward (Taqman): Hs01060665_g1 (Thermo Fisher Scientific)

The quantification cycle (Cq) values of sgN and sgE are reported as means ±SD normalized

to the control (human ACTB) of three replicates.”

Point 10: Lines 377-378: While referring to syncytia as an evolutionarily conserved cellular structure, please provide a context and an excellent reference to make your point clear.

Response 10: We thank reviewer for his suggestion. We have now added references and rephrased in the results section (Lm408-409) as follow:

“Syncytia phenomena are related to the persistence viral RNA infection and replication in SARS-CoV-2-infected patients [13, 18-19].”

Reviewer 3 Report

Ferrucci et al., present evidence that monitoring sgN transcript in COVID-19 patents can be helpful as a measure of replicating virus. They claim to have developed a Taqman-based RT-qPCR assay to detect sgN transcript along with other genes of COVID-19. This research group has published interesting articles earlier on COVID-19 (see Refs 5 and 11). However, I find the presentation of this manuscript is unclear in many areas. My primary concern is the Materials and Method section, which is inadequately described. I have raised several questions that should improve clarity and addresses the deficiencies.

General comments:

  1. I find inconsistencies in the use of PCR terminology. For example, the qPCR threshold cycle is mentioned as CT, Ct, or Cq at different regions. Better to be consistent by using Cq, which is now widely accepted.
  2. In bioinformatics terms, homology means origin from a common ancestor. Thus, a gene can be either homologous or not homologous (an analogy: a woman either pregnant or not pregnant). Thus, 80% and 50% homology as mentioned (e.g., Line 52, Line 360 ) is incorrect and requires revisions.
  3. The respiratory system is primarily affected by COVID-19. What is the rationale for selecting HET-293T cells derived from the human embryonic kidney for the transfection study?

Specific questions:

Materials and Method section:

  1. Where did you get HEK-293T cells? What are the culture conditions such as seeding density, CO2, etc? Where did you synthesize 2’-O-Methyl RNA oligos, and who designed these oligos?
  2. Line 538: Please be specific about the biological sample you used to isolate COVID-19 from the patient samples? Was it a nasopharyngeal swab?
  3. Lines 551-552: What qPCR program was used? It is not clear why did you present 25° C for 2 min……....etc?
  4. Did you multiplex SgN, E, and ACE2 genes in the Taqman assay? Where is the probe sequence for sgN?
  5. It is unclear why the human housekeeping gene ACTB was used as a normalizer and to normalize which gene?
  6. Why was the human RNase P gene used? Where are the primers of these genes?

Lines 377-378: While referring to syncytia as an evolutionarily conserved cellular structure, please provide a context and an excellent reference to make your point clear.

In summary, I find this manuscript requires major revision before it is considered for publication.

Author Response

NO comment from Reviewer n.3

Round 2

Reviewer 3 Report

Accept